# The anti-sigma factor RsrA responds to oxidative stress by reburying its hydrophobic core

Karthik V. Rajasekar[1], Konrad Zdanowski[2,†], Jun Yan[3], Jonathan T.S. Hopper[3], Marie-Louise R. Francis[1], Colin Seepersad[1], Connor Sharp[1], Ludovic Pecqueur[4,†], Jörn M. Werner[4], Carol V. Robinson[3], Shabaz Mohammed[1,3], Jennifer R. Potts[2] & Colin Kleanthous[1]

Redox-regulated effector systems that counteract oxidative stress are essential for all forms of life. Here we uncover a new paradigm for sensing oxidative stress centred on the hydrophobic core of a sensor protein. RsrA is an archetypal zinc-binding anti-sigma factor that responds to disulfide stress in the cytoplasm of Actinobacteria. We show that RsrA utilizes its hydrophobic core to bind the sigma factor $\sigma^R$ preventing its association with RNA polymerase, and that zinc plays a central role in maintaining this high-affinity complex. Oxidation of RsrA is limited by the rate of zinc release, which weakens the RsrA–$\sigma^R$ complex by accelerating its dissociation. The subsequent trigger disulfide, formed between specific combinations of RsrA's three zinc-binding cysteines, precipitates structural collapse to a compact state where all $\sigma^R$-binding residues are sequestered back into its hydrophobic core, releasing $\sigma^R$ to activate transcription of anti-oxidant genes.

[1] Department of Biochemistry, University of Oxford, South Parks Road, Oxford OX1 3QU, UK. [2] Department of Biology, University of York, York YO10 5DD, UK. [3] Chemistry Research Laboratory, Department of Chemistry, University of Oxford, 12 Mansfield Road, Oxford OX1 3TA, UK. [4] School of Biological Sciences, University of Southampton, Bassett Crescent East, Southampton SO16 7PX, UK. † Present addresses: Institute of Chemistry, University of Natural Sciences and Humanities, 3 Maja 54, 08-110 Siedlce, Poland, and Institute of Biochemistry and Biophysics Polish Academy of Sciences, Pawinskiego 5A, 02-106 Warsaw, Poland. (K.Z.); Chemistry of Biological Processes, Collège de France, 11 place Marcelin Berthelot, 75231 Paris, France. (L.P.). Correspondence and requests for materials should be addressed to C.K. (email: colin.kleanthous@bioch.ox.ac.uk).

All organisms must contend with the toxic effects of reactive oxygen species (ROS), which include superoxide anion ($O_2^-$), hydrogen peroxide ($H_2O_2$) and the hydroxyl radical (OH•)[1,2], that covalently damage proteins, lipids and DNA[3]. ROS are by-products of aerobic metabolism, which in mammals are implicated in the ageing process and diseases such as type 2 diabetes[4]. To minimize the build-up of disulfide bonds, one of the toxic consequences of ROS, organisms maintain a reducing cytoplasm through the production of millimolar concentrations of small-molecule reducing agents such as glutathione[5]. A second line of defence comprises detoxification enzymes that decompose ROS and the glutaredoxin/thioredoxin system of redox proteins that reduce cytoplasmic disulfide bonds[6]. Oxidative stress sensor proteins that lead to the activation of anti-oxidant genes form a third line of defence for maintaining redox homeostasis[7]. Sensor proteins are typically transcription factors or transcription factor inhibitors that contain reactive cysteines or metal centres that are directly modified by ROS[8,9]. Here we focus on the disulfide stress sensor protein RsrA from *Streptomyces coelicolor*, which, in its resting state, blocks binding of the sigma factor $\sigma^R$ to RNA polymerase (Fig. 1)[10,11]. RsrA is a zinc-binding anti-sigma factor (ZAS) protein, the prototypical member of a large family of inhibitors of extracytoplasmic function (ECF) sigma factors that regulate bacterial responses to diverse stresses[12]. As yet, no molecular mechanism has been described for the stress-induced inactivation of any ZAS protein. We detail the mechanism by which RsrA responds to oxidation, releasing $\sigma^R$ to mount the cellular anti-oxidant response.

ZAS proteins were originally identified by their $His_{XXX}Cys_{XX}Cys$ sequence motifs[11]. They share <30% sequence identity, but are readily identified in bacterial genomes by their genomic location, downstream of an ECF (group IV) sigma factor[13]. ZAS proteins are further sub-divided by the identity of the fourth zinc coordination site, which is either a cysteine or histidine residue 23–26 amino acids N-terminal to the $His_{XXX}Cys_{XX}Cys$ motif (hereafter these two types of ZAS motifs are denoted as CHCC or HHCC, respectively), and if they contain an additional domain or transmembrane region. RsrA is a soluble, single domain, CHCC-type ZAS motif protein, while its paralogue ChrR[14] from the photosynthetic bacterium *Rhodobacter sphaeroides* is a HHCC-type ZAS motif protein, which has an additional cupin-like domain.

ZAS proteins respond to different stimuli, inducing them to release their cognate sigma factor to activate regulons that respond to the stress[12,15]. Homologues of the RsrA–$\sigma^R$ complex are found throughout the actinomycetes, including *Mycobacterium tuberculosis*, where the system has been shown to be important for pathogenesis[16], and *Corynebacterium diphtheriae*. In the case of *S. coelicolor*, $\sigma^R$ is a global transcriptional regulator, activating a regulon of >100 genes that includes anti-oxidant genes (Fig. 1)[17]. ChrR by contrast senses singlet oxygen stress, a toxic ROS by-product of photosynthesis[18,19]. Release of its sigma factor, $\sigma^E$ (also known as RpoE), results in the increased production of carotenoids that quench the singlet oxygen radical[20]. RsiW from *Bacillus subtilis* is a membrane-bound ZAS protein that is proteolytically degraded following cell envelope stress through the action of antibiotics such as vancomycin, releasing its ECF sigma factor to activate a regulon for the detoxification of and protection against antimicrobials[21]. Structures for two ZAS proteins bound to their cognate ECF sigma factors have been reported, the intact ChrR–$\sigma^E$ complex[13] and the isolated ZAS domain of RslA from *M. tuberculosis* bound to one of the two domains of $\sigma^L$ (ref. 22). No structure has yet been reported for any ZAS protein in the absence of its sigma factor or in an inactivated state following stress-induced dissociation.

RsrA is a 105 amino-acid protein that contains seven cysteines. Three of the cysteines, Cys11, Cys41 and Cys44, contribute to the CHCC ZAS motif and are essential for redox sensing *in vivo* and *in vitro*[23]. Zdanowski et al.[24] showed using extended X-ray absorption fine structure spectroscopy that all three cysteines, along with His37, also within the ZAS motif, coordinate a single zinc ion in both RsrA and the RsrA–$\sigma^R$ complex[24]. Oxidation of RsrA is known to result in the loss of zinc and formation of a degenerate trigger disulfide bond, formed between Cys11 and either of Cys41 or Cys44, which blocks $\sigma^R$ binding[25,26]. However, the involvement of the metal ion in redox sensing remains enigmatic. Here we uncover this role and its structural basis. As well as revealing a new mechanism by which an oxidative stress sensor protein responds to the changes in cellular redox status, this study also lays the foundations for understanding how ZAS proteins function as generic stress sensors.

## Results

**Stoichiometry of zinc binding to RsrA.** We first re-assessed the stoichiometry of zinc binding to the wild-type protein and a mutant in which the four non-essential cysteines (Cys3, Cys31,

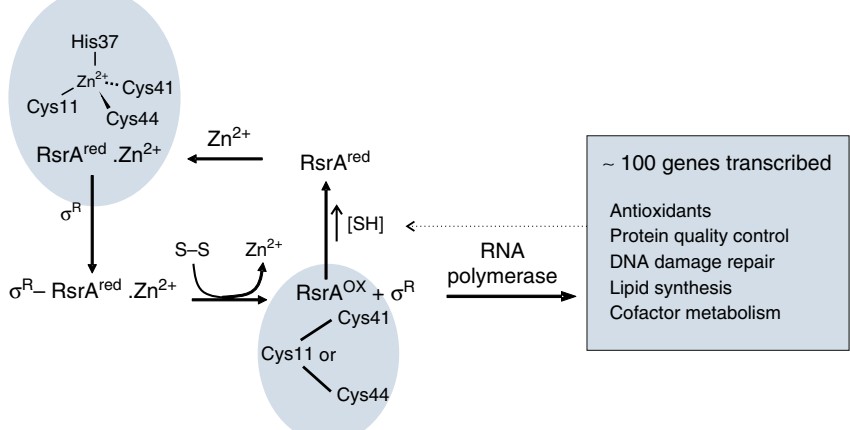

**Figure 1 | Scheme showing redox homeostasis loop for the RsrA–$\sigma^R$ complex.** The figure highlights the zinc coordination residues in reduced RsrA (RsrA$^{red}$.Zn$^{2+}$) from *Streptomyces coelicolor*. Disulfide stress results in the loss of zinc and formation of a degenerate trigger disulfide bond in RsrA$^{ox}$, formed by the same zinc-binding residues. The transcribed regulon of $\sigma^R$ includes anti-oxidant genes that re-establish redox homeostasis and the genes for *sigR* and *rsrA* (not shown), which amplify the response. Not shown is an additional layer of regulation involving a form of $\sigma^R$ with an N-terminal extension that also binds RsrA, but is rapidly degraded by proteolysis[65]. Shaded panels denote NMR structures of RsrA reported in the present work.

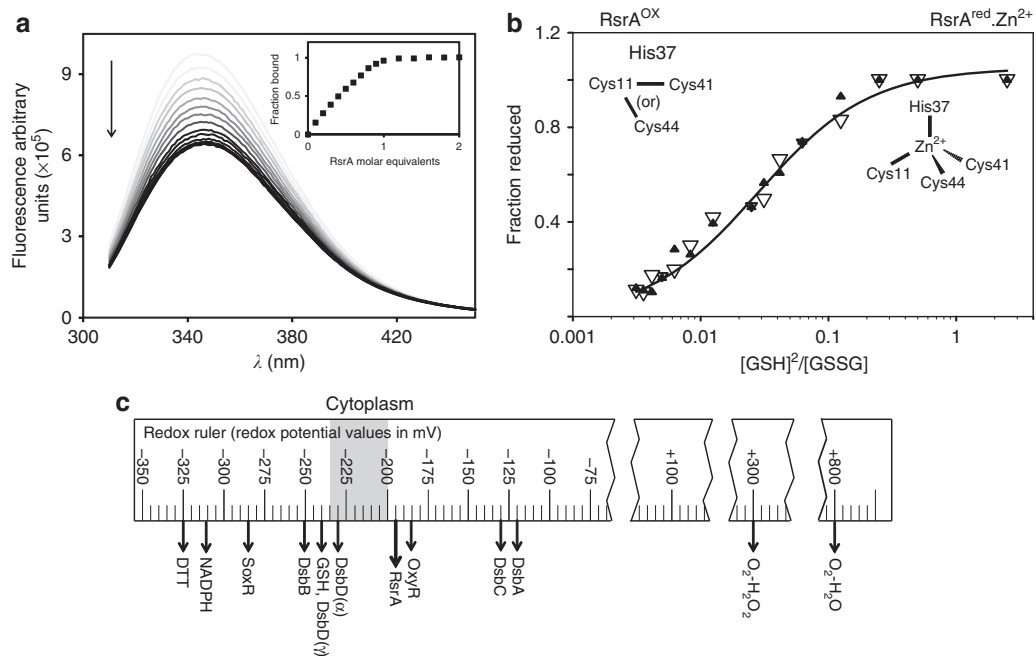

**Figure 2 | Redox potential of the RsrA$^{red}$.Zn$^{2+}$–σ$^R$ complex. (a)** RsrA$^{red}$.Zn$^{2+}$ binding to σ$^R$ monitored by intrinsic tryptophan emission fluorescence spectroscopy. Figure shows emission spectra ($\lambda_{ex}$ 295 nm) of σ$^R$ at different RsrA$^{red}$.Zn$^{2+}$ concentrations collected at 25 °C in 50 mM Tris (pH 7.5) buffer, 100 mM NaCl and 2 mM DTT. For clarity, traces are coloured with incrementally darker tone of grey for increasing RsrA concentration. Inset figure shows fraction of σ$^R$ in complex as measured by the change in fluorescence intensity at 343 nm with increasing RsrA$^{red}$.Zn$^{2+}$ concentration. Data show stoichiometric binding of RsrA$^{red}$.Zn$^{2+}$ to σ$^R$. **(b)** Redox potential of RsrA$^{red}$.Zn$^{2+}$ in complex with σ$^R$ complex relative to a redox couple of reduced/oxidized glutathione. The degree of oxidation was measured by the change in tryptophan emission fluorescence of σ$^R$ as the complex dissociates (open triangles) and by the release of Zn$^{2+}$ using the PAR assay (filled triangles). Both sets of data were fitted to the Nernst equation to determine the redox potential of the RsrA$^{red}$-Zn$^{2+}$–σ$^R$ complex as  − 193.04 ± 2.01 mV. Experiments were conducted in 50 mM Tris (pH 7.5) buffer containing 100 mM NaCl and at 25 °C. **(c)** Redox ruler showing the redox potentials of selected proteins and small molecules. Figure adapted from ref. 27. The estimated redox potential of the bacterial cytoplasm is indicated in grey.

Cys61 and Cys62) were mutated to alanine (RsrA*). See the Methods section for details. Although multiple zinc ions can bind to reduced RsrA and RsrA*, only a single zinc stabilizes the protein fold (Supplementary Fig. 1) and, as detailed below, modulates redox activity. We refer to this form as RsrA$^{red}$.Zn$^{2+}$.

**Redox potential of RsrA$^{red}$.Zn$^{2+}$.** A key question for an oxidative stress sensor is its redox potential, as this will govern its reactivity towards oxidants. No such measurements have been reported for any ZAS protein. We therefore determined the redox potential for RsrA$^{red}$.Zn$^{2+}$ in complex with σ$^R$ with reference to a glutathione redox couple. The status of the complex in these experiments was monitored by tryptophan emission fluorescence spectroscopy, where we exploited a significant change in σ$^R$ fluorescence that occurs on forming its complex with RsrA$^{red}$.Zn$^{2+}$ (Fig. 2a). The oxidation status of RsrA$^{red}$.Zn$^{2+}$ was determined spectrophotometrically by the stoichiometric release of zinc using 4-(2-pyridylazo) resorcinol (PAR; see Methods). The two data sets were in excellent agreement (Fig. 2b) and showed the redox potential for RsrA$^{red}$-Zn$^{2+}$ in complex with σ$^R$ to be  − 193.04 ± 2.01 mV. Our data show that the redox potential of RsrA$^{red}$.Zn$^{2+}$ is ideally poised to act as a redox sensor. Its redox potential is close to that estimated for the bacterial cytoplasm (Fig. 2c)[27], rendering it sensitive to small changes in the redox status of the cell.

**Role of zinc in modulating σ$^R$ binding and oxidation of RsrA.** Using isothermal titration calorimetry (ITC), we determined that RsrA$^{red}$.Zn$^{2+}$ binds σ$^R$ with sub-nanomolar affinity (Fig. 3a),

whereas RsrA devoid of zinc-bound σ$^R$ 100-fold more weakly (Fig. 3b). Zinc-associated RsrA*-bound σ$^R$ with a similar affinity to wild-type RsrA, while mutation of any zinc-binding cysteine residue decreased σ$^R$ binding by >100-fold (Supplementary Table 1). We conclude that the RsrA$^{red}$.Zn$^{2+}$–σ$^R$ complex has a much higher affinity than previously reported[11,28], which is dependent on zinc being bound at the ZAS motif and explains why zinc limitation activates the σ$^R$ regulon in *S. coelicolor*[29].

We next determined the kinetic basis for zinc stabilization of the complex. Stopped-flow tryptophan fluorescence under pseudo-first-order conditions showed that the association rate constant for the RsrA$^{red}$–σ$^R$ complex was only marginally affected by bound zinc, in contrast to the dissociation rate constant that was accelerated 400-fold when zinc was removed (Fig. 3c,d). Importantly, the kinetically derived $K_d$ for the RsrA$^{red}$.Zn$^{2+}$–σ$^R$ complex at 35 °C closely matched that obtained by ITC (Supplementary Table 2), demonstrating that the kinetic mechanism can simply be described by single association and dissociation rate constants. The kinetically derived $K_d$ at 25 °C (for which a value could not be obtained by ITC) was approximately twofold lower than that at 35 °C, with zinc having very similar effects on the kinetics of binding (Supplementary Table 2). We conclude that the single ZAS motif zinc ion of RsrA stabilizes the high-affinity complex with σ$^R$ by slowing the dissociation rate of the complex.

RsrA is thought to be primarily a sensor of deleterious disulfide bond formation within the *S. coelicolor* cytoplasm, since oxidation by the thiol-specific oxidizing agent diamide is a stronger inducer of σ$^R$-dependent promoters than hydrogen peroxide[10]. We therefore determined pre-steady-state oxidation rates of

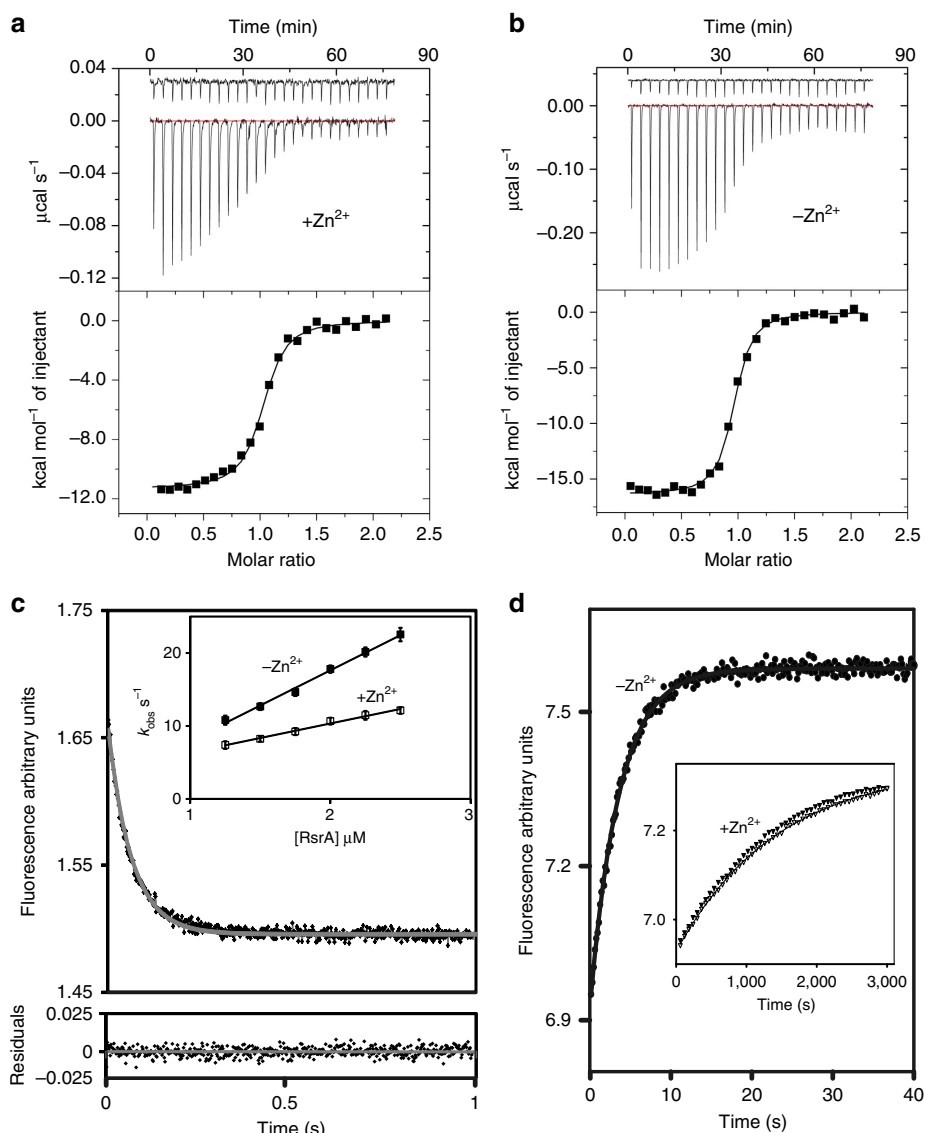

**Figure 3 | Zinc slows the dissociation rate of the high-affinity RsrA$^{red}$.Zn$^{2+}$-$\sigma^R$ complex.** Conditions used for all experiments were 50 mM Tris (pH 7.5) buffer containing 100 mM NaCl and 2 mM DTT. Temperature was either 35 °C (**a,b**) or 25 °C (**c,d**). (**a**) Competition ITC data for RsrA$^{red}$.Zn$^{2+}$ (100 µM) binding $\sigma^R$ (10 µM, cell concentration) in the presence of RsrA* Cys11Ser Cys44Ser (50 µM). RsrA* Cys11Ser Cys44Ser binds $\sigma^R$ with a weaker affinity than wild-type RsrA (Supplementary Table 1). Control data show titration of RsrA$^{red}$.Zn$^{2+}$ into buffer containing 50 µM RsrA* Cys11Ser Cys44Ser. Fitted parameters for a competitive binding site model from three independent measurements were $N = 1.01 \pm 0.05$, $K_d = 0.78 \pm 0.034$ nM, $\Delta H = -23.05 \pm 1.10$ kcal mol$^{-1}$, $\Delta S = -38.70 \pm 3.10$ cal mol$^{-1}$ deg$^{-1}$. (**b**) Direct ITC data for RsrA$^{red}$ (100 µM) binding $\sigma^R$ (10 µM) in the absence of zinc. Control data show titration of RsrA$^{red}$ into buffer. Fitted parameters for a single site-binding model were $N = 0.97 \pm 0.004$, $K_d = 79.3 \pm 4.2$ nM, $\Delta H = -16.26 \pm 0.13$ kcal mol$^{-1}$, $\Delta S = -20.3 \pm 4.34$ cal mol$^{-1}$ deg$^{-1}$. (**c**) Main panel: tryptophan emission fluorescence stopped-flow association data for RsrA$^{red}$.Zn$^{2+}$ binding $\sigma^R$ under pseudo-first-order conditions ($\lambda_{ex}$, 295 nm). Residuals to the fit of a single exponential are shown below the panel. Inset: pseudo-first-order plot of observed rates ($k_{obs}$) as a function of RsrA concentration in the presence and absence of stoichiometric zinc (with associated error bars). See Supplementary Table 2 for derived values of $k_{on}$. Error bars are within the data point symbols. (**d**) Dissociation of the RsrA$^{red}$-$\sigma^R$ complex measured by competition stopped-flow in which a 10-fold excess of $\sigma^R$ Trp88Ile Trp119Ile was used to displace wild-type $\sigma^R$ (Methods). Main panel shows data for the RsrA$^{red}$-$\sigma^R$ complex in the absence of bound zinc. Inset: dissociation data for the RsrA$^{red}$-$\sigma^R$ complex in the presence of 1 and 3 equiv. of zinc (open and filled triangles, respectively). See Supplementary Table 2 for values of $k_{off}$.

RsrA$^{red}$.Zn$^{2+}$ in complex with $\sigma^R$ to probe the kinetic basis for oxidation, initially using diamide to induce disulfide bond formation within RsrA (Fig. 4a,b; see Methods). We developed a stopped-flow spectrophotometric assay to follow RsrA$^{red}$.Zn$^{2+}$ oxidation, albeit indirectly, by exploiting the absorption changes of diamide on reduction to hydrazine (Supplementary Fig. 2a)[30]. Zinc release was monitored using the PAR assay (Fig. 4a). These data showed that the bimolecular rate constant for zinc release is the same as that of diamide reduction ($\sim 190$ M$^{-1}$ s$^{-1}$),

suggesting that zinc release limits the rate of oxidation. This was confirmed by removing zinc from the protein, which accelerated the oxidation rate fivefold (Fig. 4a). Identical results were obtained for RsrA$^{red}$.Zn$^{2+}$ in the absence of $\sigma^R$, demonstrating that complex formation does not influence the kinetics of oxidation (Supplementary Fig. 2c). Moreover, the release of zinc with increasing diamide concentration showed the oxidant formed a weak intermediate complex ($K_1 \sim 0.7$ mM; Fig. 4b), most likely the sulfenyl hydrazine (Supplementary

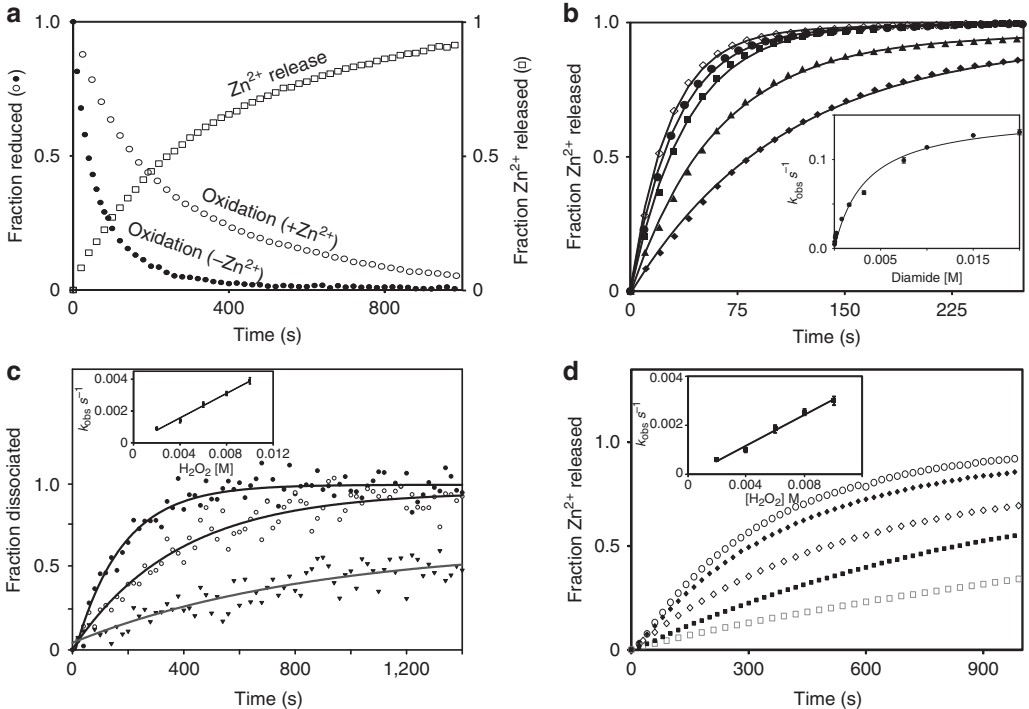

**Figure 4 | Zinc release is the rate-limiting step in RsrA oxidation.** Experiments were conducted at 25 °C in 50 mM Tris (pH 7.5) buffer containing 100 mM NaCl. (**a**) Oxidation of the RsrA–σ$^R$ complex on treatment with diamide under second-order conditions (25 μM). The fraction of reduced RsrA was determined by the change in diamide absorbance at 320 nm (Supplementary Fig. 2a) in the presence and absence of bound zinc. Zinc release was monitored at 500 nm using the PAR assay. The two methods showed good agreement for the bimolecular rate constant for diamide-induced oxidation of RsrA$^{red}$.Zn$^{2+}$ (183 ± 6 and 195 ± 11 M$^{-1}$s$^{-1}$, respectively). (**b**) Zinc release from the RsrA$^{red}$.Zn$^{2+}$–σ$^R$ complex (2 μM) on treatment with increasing concentrations of diamide under pseudo-first-order conditions monitored by the PAR assay; 25 μM (filled diamonds), 50, 100, 150 and 200 μM (open diamonds). Inset shows variation of $k_{obs}$ with diamide concentration (with associated error bars). Data were fitted to the Michaelis–Menten equation, with fitted parameters of $K_1 = 0.7$ mM and $k_2 = 0.15$ s$^{-1}$ (Supplementary Fig. 2b). The corresponding bimolecular rate constant ($k_2/K_1 = 214$ M$^{-1}$s$^{-1}$) is in reasonable agreement with values obtained in **a**. (**c**) Oxidation-induced dissociation of the RsrA$^{red}$.Zn$^{2+}$–σ$^R$ complex (2 μM) monitored by tryptophan fluorescence on treatment with H$_2$O$_2$ under pseudo-first-order conditions. Three H$_2$O$_2$ concentrations are shown as follows: 1 mM (triangles), 2 mM (open circles) and 6 mM (closed circles). Inset, pseudo-first-order plot (with associated error bars) from which the bimolecular rate constant for H$_2$O$_2$-induced dissociation of the complex was obtained (0.39 ± 0.08 M$^{-1}$s$^{-1}$). (**d**) Zinc release from the RsrA$^{red}$.Zn$^{2+}$–σ$^R$ complex (2 μM) on treatment with increasing concentrations of H$_2$O$_2$ (2–10 mM) under pseudo-first-order conditions. Inset: pseudo-first-order plot (with associated error bars) from which the bimolecular rate constant for the H$_2$O$_2$-induced zinc dissociation was obtained (0.32 ± 0.06 M$^{-1}$s$^{-1}$).

Fig. 2a,b), before formation of the trigger disulfide bond. Importantly, the kinetic analysis showed that the first-order rate constant ($k_2$) for decomposition of this intermediate complex is 0.15 s$^{-1}$ at 25 °C. This rate approaches the intrinsic dissociation rate of the RsrA–σ$^R$ complex in the absence of zinc under the same conditions (∼0.3 s$^{-1}$; Supplementary Table 2), which is consistent with oxidation increasing the dissociation rate of the complex by driving out bound zinc.

The absorbance of diamide precluded monitoring dissociation of the RsrA–σ$^R$ complex directly by fluorescence spectroscopy. This was however possible using H$_2$O$_2$ as an oxidant, where the release of zinc could also be monitored (Fig. 4c,d). Although the bimolecular rate constant for oxidation by H$_2$O$_2$ was 500-fold slower than that of diamide (emphasizing that RsrA is a sensor of disulfide rather than peroxide stress) here again zinc release was rate limiting for oxidation and complex dissociation. Importantly, excess zinc had no effect on the rates of oxidation either by diamide or H$_2$O$_2$ (Supplementary Fig. 3b,c), demonstrating that additional zinc ions beyond that bound in the ZAS motif play no role in redox sensing by RsrA. In conclusion, our kinetic data demonstrate that stoichiometric zinc release is the rate-limiting step for RsrA oxidation by different oxidants, which leads to accelerated dissociation of the RsrA–σ$^R$ complex and formation of the trigger disulfide bond (a full kinetic scheme is shown in Supplementary Fig. 3d).

**Cys11 is essential for redox sensing in RsrA.** Heo *et al.*[31] have suggested that the differences in redox sensitivity of different ZAS proteins are due to the differences in electronegative residues and binding of alternative zinc ions. However, as our data above illustrate, additional zinc ions play little or no role in the redox-sensing ability of RsrA. A simpler explanation for whether ZAS proteins react to disulfide stress is whether the N-terminal zinc-coordinating residue is a cysteine, as in the case of RsrA where Cys11 forms the trigger disulfide with either Cys41 or Cys44. HHCC-type ZAS proteins such as ChrR and RsiW do not have this additional cysteine and do not sense disulfide stress. This hypothesis is confounded however by the recent study of Thakur *et al.*[22] who reported that on oxidation with hydrogen peroxide, the HHCC ZAS protein RslA from *M. tuberculosis* forms a disulfide bond, expelling the bound zinc and increasing the dissociation rate of the RslA–σ$^L$ complex[22]. We therefore set out to determine how different zinc ligation chemistries influence oxidative stress sensing, using the RsrA–σ$^R$ complex as a model. We generated four cysteine mutants in RsrA*: RsrA* in which each of the zinc-coordinating cysteines was individually mutated to histidine and RsrA* Cys11His Cys41His Cys44His, in which all the coordinating cysteines were simultaneously replaced with histidine. The latter mutant was used as a non-oxidative control. The Cys-to-His mutations all bound σ$^R$ with lower

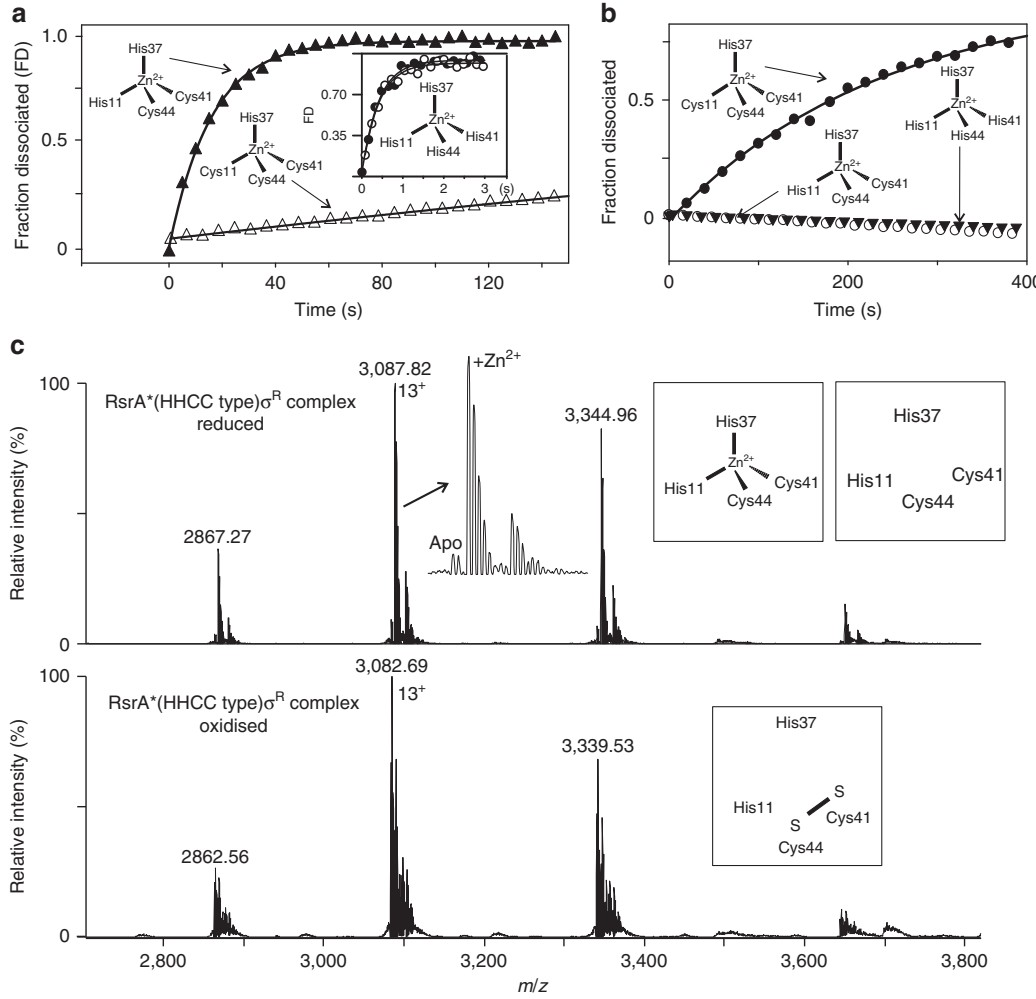

**Figure 5 | Cys11 is essential for redox sensing in RsrA.** Zinc ligation for each RsrA construct used is shown as a schematic. (**a**) Comparing the intrinsic dissociation rates of $\sigma^R$ complexes with RsrA\*$^{red}$.Zn$^{2+}$ (open triangles; $k_{off} = 0.0027\,s^{-1}$) and RsrA\*$^{red}$.Zn$^{2+}$ Cys11His (closed triangles; $k_{off} = 0.064\,s^{-1}$) obtained by competition stopped-flow tryptophan emission fluorescence using a 10-fold excess of $\sigma^R$ Trp88Ile Trp119Ile. All experiments were conducted at 25 °C in 50 mM Tris (pH 7.5) buffer containing 100 mM NaCl. Inset: dissociation data in the presence (closed circles) and absence (open circles) of zinc for $\sigma^R$ in complex with triple mutant RsrA\* Cys11His Cys41His Cys44His ($k_{off} \sim 2.3\,s^{-1}$). (**b**) Comparison of H$_2$O$_2$ (10 mM)-induced dissociation of the RsrA\*$^{red}$.Zn$^{2+}$–$\sigma^R$ complex (closed circles) with RsrA\*$^{red}$.Zn$^{2+}$ Cys11His–$\sigma^R$ (open circles) and RsrA\*$^{red}$.Zn$^{2+}$ Cys11His Cys41His Cys44His (closed triangles) complexes. Data for complexes of RsrA\* Cys41His and RsrA\* Cys44His, shown in Supplementary Fig. 4, were essentially identical to RsrA\*. The absence of Cys11 renders RsrA insensitive to oxidation-induced dissociation from its complex with $\sigma^R$. (**c**) Upper panel: native state mass spectrometry data for RsrA\*$^{red}$.Zn$^{2+}$ Cys11His in complex with $\sigma^R$ showing the predominance of the zinc-bound species (theoretical/observed mass, 40,128.08/40,127.54 ± 0.12 Da, respectively). The minor species was the reduced complex without zinc bound (theoretical/observed mass, 40,064.72/40,064.41 ± 0.66 Da, respectively). Lower panel: the same complex following treatment with 10 mM H$_2$O$_2$. Although RsrA\* Cys11His remains in complex with $\sigma^R$ the metal ion has dissociated and the remaining cysteines (Cys41 and Cys44) have formed a disulfide bond, as deduced by the $\sim 2$ Da reduction in mass (theoretical/observed mass, 40,062.70/40,061.96 ± 0.28 Da, respectively).

affinity than wild-type RsrA, the mutations affecting primarily the dissociation rate of the RsrA\*–$\sigma^R$ complex (Fig. 5a; Supplementary Table 1).

We next challenged RsrA\* and all the cysteine mutants with 10 mM H$_2$O$_2$ and followed the kinetics of oxidation-induced dissociation of their complexes with $\sigma^R$ by fluorescence spectroscopy (Fig. 5b). In contrast to RsrA\*, which had similar oxidation-induced dissociation kinetics to wild-type RsrA, neither RsrA\* Cys11His nor the triple mutant, RsrA\* Cys11His Cys41His Cys44His, oxidatively dissociated when challenged with H$_2$O$_2$. However, the single mutants RsrA\* Cys41His and RsrA\* Cys44His both exhibited identical oxidation-induced dissociation kinetics to RsrA\*, consistent with the degeneracy of the trigger disulfide bond (Supplementary Fig. 4).

We further analysed RsrA\* Cys11His (equivalent to a HHCC motif ZAS protein) in complex with $\sigma^R$ using high-resolution native state nanoelectrospray mass spectrometry to determine the oxidation state of this HHCC-type ZAS protein (Fig. 5c). Under reducing conditions, the RsrA\* Cys11His–$\sigma^R$ complex bound 1 equiv. of zinc with only a small fraction of apo-complex present. Upon H$_2$O$_2$-induced oxidation of the complex, a mass shift to lower $m/z$ indicated both the loss of zinc and formation of a disulfide bond between Cys41 and Cys44. However, this did not result in oxidation-induced dissociation of the complex (Fig. 5b,c). These data suggest that oxidative dissociation of a ZAS protein from its target sigma factor requires an N-terminal zinc-coordinating cysteine residue within the ZAS motif (CHCC).

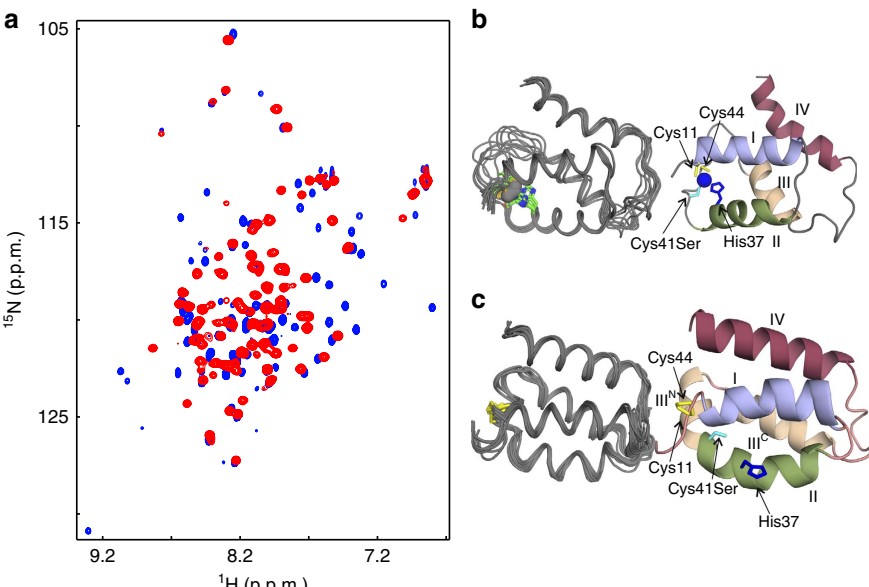

**Figure 6 | Solution structures of RsrA$^{red}$.Zn$^{2+}$ and RsrA$^{ox}$. (a)** Comparison of $^{1}$H-$^{15}$N-HSQC spectra of reduced RsrA* Cys41Ser (RsrA$^{red}$-Zn$^{2+}$, blue peaks), in 20 mM Tris buffer (pH 7.1) containing 5 mM DTT and 2 mM ZnCl$_2$, with RsrA$^{ox}$ (red peaks), which is the same protein in the same buffer but in the absence of reductant and metal ions. **(b)** Overlay of the 10 lowest-energy structures for RsrA$^{red}$-Zn$^{2+}$ (residues Glu8–Gln86; left-hand figure) and a ribbon diagram of the lowest-energy structure (right-hand figure), showing the location of the zinc-binding residues (His37, blue; Cys41Ser, cyan; Cys11 and Cys44, yellow) and the zinc atom (blue). **(c)** Overlay of the 10 lowest-energy structures for RsrA$^{ox}$ (residues Glu8–Gln86; left-hand figure) and a ribbon diagram of the lowest-energy structure (right-hand figure), showing the location of the disulfide bond and disruption of the metal site following oxidation (residues and helices are coloured as in **b**).

**RsrA exposes its hydrophobic core to bind σ$^R$.** To understand how RsrA associates with σ$^R$, we determined the structure of zinc-bound RsrA in its reduced state (RsrA$^{red}$.Zn$^{2+}$) and compared this with a homology model of the RsrA$^{red}$.Zn$^{2+}$–σ$^R$ complex. A modelling approach was employed because repeated attempts at crystallization of the RsrA$^{red}$.Zn$^{2+}$–σ$^R$ complex failed to yield diffracting crystals and solution spectra of $^2$H, $^{13}$C, $^{15}$N-labelled RsrA$^{red}$.Zn$^{2+}$–σ$^R$ complex were poorly resolved. The model was based on previous structures of anti-sigma factor–sigma factor complexes and constrained by bifunctional lysine-specific crosslinking data (Supplementary Fig. 5). Following testing of RsrA mutants for optimal spectral resolution (including RsrA*), the nuclear magnetic resonance (NMR) solution structure of RsrA$^{red}$.Zn$^{2+}$ was obtained using the mutant RsrA* Cys41Ser (which contains both Cys11 and Cys44) bound to 1 equiv. of zinc. As for the wild-type protein, $^1$H-$^{15}$N-HSQC (heteronuclear single quantum coherence) NMR spectra showed that this mutant required stoichiometric zinc for its stabilization (Supplementary Fig. 6a). Although RsrA* Cys41Ser likely binds zinc more weakly than wild-type RsrA, leading to weakened σ$^R$ binding (Supplementary Table 1), at the protein and zinc concentrations used for NMR structure determination (∼mM) the protein is folded and bound to zinc. Only residues 45–47 in RsrA* Cys41Ser could not be assigned by heteronuclear NMR experiments. In the final structure, the N and C termini, a loop between residues 63 and 72, and residues 42 and 50 were poorly resolved, all other residues (8–86) were well defined. Heavy-atom root mean square deviations (r.m.s.d.'s) for backbone atoms of the secondary structure elements in the 10 overlays of RsrA$^{red}$.Zn$^{2+}$ shown in Fig. 6b was 0.48 Å. NMR structure statistics are shown in Table 1.

RsrA$^{red}$.Zn$^{2+}$ (Fig. 6b) forms a loosely packed four-helix bundle composed of two sets of roughly parallel helices (I–II, residues 11–23 and 29–39; and III–IV, residues 51–60 and 71–84) connected by loops. The two N-terminal helices are tilted ∼45° relative to the C-terminal helices. His37 of the conserved ZAS metal-binding motif is presented from the C-terminal end of helix II, while Cys41Ser and Cys44 are part of the long loop connecting helices II and III. Cys11, the fourth metal ligand, is at the N-terminal end of helix I. The co-localization of the four metal ligands was confirmed by the observation of inter-residue nuclear overhauser effect (NOEs); distance restraints were used during initial structure calculations and restraints specifying the tetrahedral Zn$^{2+}$ ligation geometry were introduced in the latter stages of refinement (Methods).

On binding σ$^R$ RsrA$^{red}$.Zn$^{2+}$ adopts a characteristic anti-sigma-binding domain (ASD) fold, which was first described for the ChrR–σ$^E$ complex[13] (Fig. 7b). The two key features of the modelled complex are the binding of RsrA$^{red}$.Zn$^{2+}$ between the two domains of σ$^R$ (σ$^2$ and σ$^4$) and the embrace of the sigma factor by the C-terminal helix (helix IV) of RsrA$^{red}$.Zn$^{2+}$. Our crosslinking data suggest however that helix IV does not adopt a single conformation as in the ChrR–σ$^E$ complex, but can likely contact both σ$^2$ and σ$^4$ domains of σ$^R$ (Supplementary Fig. 5c). For the purposes of the following analysis, we focus only on the form of the complex in which helix IV is docked onto the σ$^2$ domain.

Comparison of the structure of free RsrA$^{red}$.Zn$^{2+}$ with the σ$^R$-bound state reveals significant structural reorganization of the anti-sigma factor while maintaining its zinc coordination geometry (compare Fig. 7a,b). (1) The four-helix bundle structure of RsrA$^{red}$.Zn$^{2+}$ converts to the three-helix ASD fold. This involves helix III changing its orientation by ∼90°, helix II by ∼30° relative to helix I and helix IV dissociating from the main body of the protein. (2) Helix III in RsrA$^{red}$.Zn$^{2+}$ approximately doubles in length. The residues comprising this extension were originally the long loop connecting helices II and III in RsrA$^{red}$.Zn$^{2+}$. As a consequence, Cys44 of the ZAS motif becomes part of helix III, while Cys41 sits between helices II and III. The other ZAS motif residues remain within their original secondary structure elements. The orientation of the extended helix III is dictated by Cys44's role in zinc coordination, emphasizing the importance of zinc in σ$^R$ binding as it allows

**Table 1 | NMR and refinement statistics for the structures of RsrA$^{ox}$ and RsrA$^{red}$-Zn$^{2+}$.**

| | RsrA$^{ox}$ | RsrA$^{red}$-Zn$^{2+}$ |
|---|---|---|
| *NMR distance and dihedral constraints* | | |
| Distance constraints | | |
| Total NOE | 1,532 | 1,033 |
| Intra-residue | 749 | 577 |
| Inter-residue | 660 | 422 |
| Sequential ($|i-j|=1$) | 333 | 227 |
| Medium range ($1<|i-j|<4$) | 218 | 108 |
| Long range ($|i-j|>5$) | 109 | 87 |
| Hydrogen bonds | 0 | 0 |
| RDC-based restraints | 40 | 29 |
| Total dihedral angle restraints | | |
| $\phi/\psi$ | 100 | 104 |
| | | |
| *Structure statistics* | | |
| Violations | | |
| Distance constraints $>0.5$ Å | 0 | 0 |
| Dihedral angle constraints $>5°$ | 0 | 0 |
| Max. dihedral angle violation (°) | 4.7 | 4.6 |
| Max. distance constraint violation (Å) | 0.44 | 0.46 |
| Deviations from idealized geometry (mean and s.d.) | | |
| Bond lengths (Å) | $0.005 \pm 0.0001$ | $0.004 \pm 0.00007$ |
| Bond angles (°) | $0.63 \pm 0.01$ | $0.49 \pm 0.01$ |
| Average pairwise r.m.s. deviation* (Å) | | |
| Heavy | $0.84 \pm 0.11$ | $0.82 \pm 0.10$ |
| Backbone | $0.34 \pm 0.07$ | $0.40 \pm 0.14$ |
| | | |
| *Ramachandran statistics* | | |
| Residues in most favoured regions/additional | 93.5% | 97.7% |
| Residues in generously allowed regions | 5.3% | 2.1% |
| Residues in disallowed regions | 1.0% [†] | 0 |

RDC, residual dipolar coupling.
*Averaged over secondary structure of 10 lowest-energy structures.
†None were well-defined residues.

the helices of RsrA to re-organize around the metal ion. Importantly, the extended regions of helix III no longer pack against helices I and II of RsrA$^{red}$.Zn$^{2+}$ as in the free state. Exposed hydrophobic residues in the C-terminal half of helix III (Leu50, Ala53, Val54 and Leu57), which were originally part of the hydrophobic core in the free RsrA$^{red}$.Zn$^{2+}$ state, now interact with σ$^R$. Conversely, hydrophobic residues (Leu45 and Tyr48) that were part of the loop between helices II and III in the unbound state and partially solvent exposed now form part of the hydrophobic core of RsrA$^{Red}$-Zn$^{2+}$ in its σ$^R$-bound state. (3) The telescopic extension of helix III projects helix IV away from the body of RsrA allowing the anti-sigma factor to wrap around σ$^R$. (4) Hydrophobic residues within helix IV of RsrA, specifically Val75 and Leu79, which were peripheral hydrophobic core residues in RsrA$^{Red}$-Zn$^{2+}$, now make contact with the sigma factor. (5) Several of the bulky residues from RsrA's hydrophobic core which bind σ$^R$ (Val54, Leu57, Val75 and Leu79) are conserved or conservatively substituted within the ZAS protein family, suggesting that they serve similar roles in all ZAS proteins (Fig. 7d). The contact sites of these hydrophobic residues on σ$^R$ are consistent with their blocking important interactions of the sigma factor with RNA polymerase, as originally described by Campbell et al.[13]. (6) RsrA double alanine mutants (Val54Ala Leu57Ala and Val75Ala Leu79Ala) each weakened binding to σ$^R$ by $>100$-fold, consistent with their making stabilizing contacts with the sigma factor (Supplementary Table 1). In summary, the loosely packed four-helical bundle structure of RsrA$^{red}$.Zn$^{2+}$ undergoes large-scale structural remodelling on binding σ$^R$ while maintaining the same ligation chemistry of the ZAS motif zinc ion. These conformational changes open up the structure of

RsrA$^{red}$.Zn$^{2+}$ and enable its embrace of σ$^R$ using hydrophobic residues released from its hydrophobic core.

**RsrA sequesters its σ$^R$-contacting residues on oxidation.** We next determined the structure of oxidized RsrA (RsrA$^{ox}$) to understand how this blocks σ$^R$ binding. We first ascertained which of the two oxidized forms of the trigger disulfide predominate at equilibrium (Supplementary Figs 6 and 7). These experiments focused on RsrA*, which behaves as a redox sensor *in vivo*[23] and *in vitro* (Fig. 5b). RsrA* Cys11–Cys44 was found to be the most populated oxidized state. We therefore determined the solution structure of oxidized RsrA* containing the Cys11–Cys44 disulfide bond. Cys41 was mutated to serine in this construct to remove the potential for mixed disulfide bond formation (Methods). As with RsrA$^{red}$.Zn$^{2+}$, the N termini of RsrA$^{ox}$ were disordered and the loop between residues 63 and 72 poorly defined. Heavy-atom r.m.s.d.'s for the top 10 solution structures of residues 8–86 in RsrA$^{ox}$ were 0.38 Å (Fig. 6c; Table 1 for structure statistics).

$^1$H-$^{15}$N-HSQC spectra of RsrA$^{red}$.Zn$^{2+}$ and RsrA$^{ox}$ are substantially different suggestive of distinct folds (Fig. 6a). This is confirmed by the structure of RsrA$^{ox}$, which is more helical than RsrA$^{red}$.Zn$^{2+}$ (Fig. 6b,c). RsrA$^{ox}$ is also more compact than RsrA$^{red}$.Zn$^{2+}$, with 13% less solvent accessible surface area (RsrA$^{red}$-Zn$^{2+}$, $2,608 \pm 28$ Å$^2$ and RsrA$^{ox}$, $2,270 \pm 33$ Å$^2$ for residues 8–86). Helices I (residues 13–24) and II (residues 31–40) are pulled closer together by the disulfide between Cys11 and Cys44, and helices III$^C$ (50–60) and IV (70–84) reorient to become roughly parallel to those of helices I and II. An additional short helix (helix III$^N$; residues 42–48), perpendicular to the other

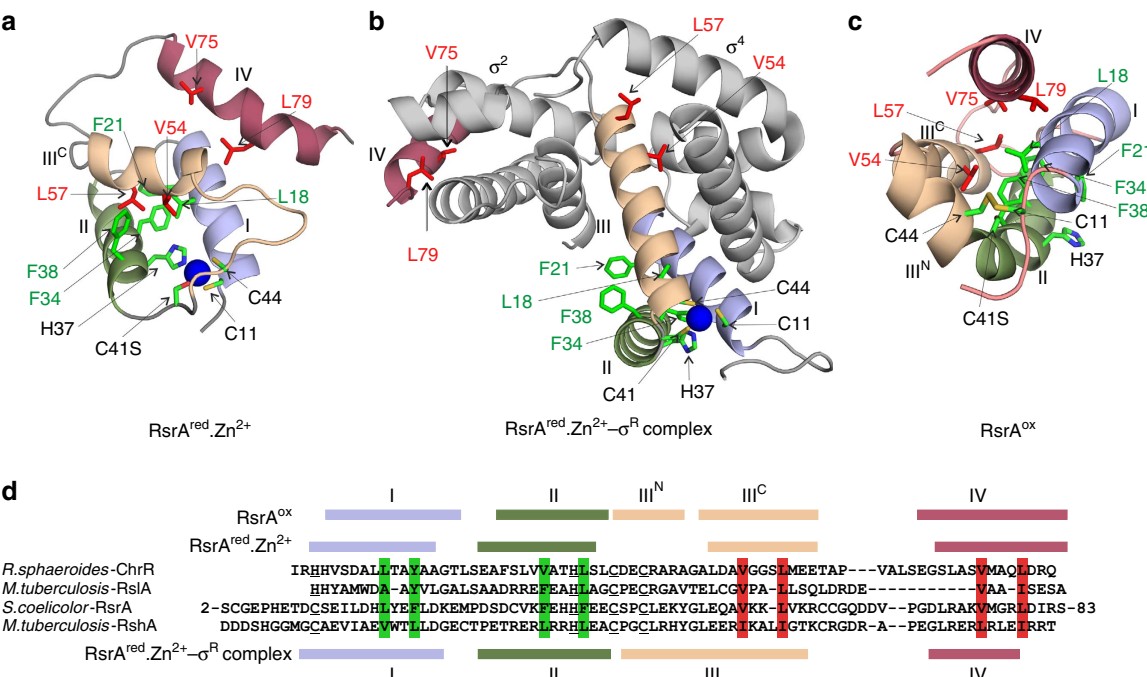

**Figure 7 | RsrA$^{red}$.Zn$^{2+}$ uses hydrophobic core residues to bind σ$^R$ that are sequestered to the RsrA$^{ox}$ interior following oxidation.** (a) Solution structure of RsrA$^{red}$.Zn$^{2+}$. RsrA helices are coloured N to C as in the sequence alignment in **d**. Zinc is shown as a blue sphere and zinc ligands coloured by atom type. Conserved or conservatively substituted hydrophobic residues that contribute to RsrA's hydrophobic core in all three of its structural states (**a–c**) are coloured green, while those that also interact with σ$^R$ are coloured red. (**b**) Homology model of the RsrA$^{red}$.Zn$^{2+}$–σ$^R$ complex validated by homo-bifunctional lysine-specific crosslinking (Supplementary Fig. 5). The structure of RsrA$^{red}$.Zn$^{2+}$ changes markedly to embrace σ$^R$. (**c**) Solution structure of RsrA$^{ox}$ where the trigger disulfide is formed between residues Cys11 and Cys44, expelling bound zinc and repacking the hydrophobic core. (**d**) Sequence alignment of RsrA and other ZAS proteins (ChrR, RshA and RslA). Zinc ligands in each protein are underlined. Helices in all three structural forms of RsrA are coloured as in **a–c**. Vertical green shading shows conserved hydrophobic residues that contribute to the hydrophobic core of RsrA in all three structural states (RsrA$^{red}$.Zn$^{2+}$, RsrA$^{red}$.Zn$^{2+}$–σ$^R$ complex and RsrA$^{ox}$). Vertical red shading shows conserved hydrophobic residues in RsrA that contribute to the hydrophobic cores of RsrA$^{red}$.Zn$^{2+}$ and RsrA$^{ox}$, but also contribute to the protein–protein interface in the RsrA$^{red}$.Zn$^{2+}$–σ$^R$ complex.

helices and stabilized by the disulfide, takes the place of the ZAS metal-binding site. Indeed, residues comprising helix III in RsrA$^{ox}$ are equivalent to those in the σ$^R$-bound state of RsrA, but now the helix is broken into two segments, a 90° one-residue turn connecting helices III$^N$ and III$^C$ (Fig. 7b–d).

These changes have three major consequences. (1) The zinc-binding site of RsrA$^{red}$.Zn$^{2+}$ is obliterated; this is most readily appreciated by the distance between NE2 atom of His37 and the S atom of Cys11 (15 Å; Fig. 6b,c). (2) The movement of ZAS ligands away from the metal-binding site is brought about by a change in register of helix II relative to the other helices due to a rotation around the helix axis. (3) The trigger disulfide bond constrains the orientations of helices I and II along with III$^N$ and III$^C$, resulting in wholesale repacking of its hydrophobic core.

The remodelling of the hydrophobic core is exemplified by changes associated with the reorientation of His37 (Fig. 7a–c). In RsrA$^{red}$.Zn$^{2+}$, zinc ligation by His37 necessitates rotation of Phe38 out of the hydrophobic core. In RsrA$^{ox}$, Phe38 (helix II) interacts with Val54 (helix III$^C$), and Leu18 and Phe21 (helix I) within the hydrophobic core of the protein, which keeps helix III$^C$ packed onto helices I and II, and so blocking RsrA's ability to interact with σ$^R$. Moreover, Val75 in helix IV also forms hydrophobic contacts with residues in RsrA$^{ox}$ (Leu18, Val54 and Leu57). Hence, formation of the trigger disulfide between Cys11 and Cys44 propagates collapse of the σ$^R$-bound form of RsrA, blocking the structural reorganization required for extension of helix III and release of helix IV. The consequence of these structural changes is that σ$^R$-contacting residues are sequestered back into RsrA's hydrophobic core.

The structure of RsrA$^{ox}$ explains why the Cys41–Cys44 disulfide does not cause dissociation of the complex. Whereas a disulfide between Cys11 and Cys44 pins helices I and III$^N$ together, stabilizing the hydrophobic core and sequestering key hydrophobic residues away from σ$^R$, a disulfide between Cys41 and Cys44 places no constraints on helix I. Hence, RsrA with the Cys41–Cys44 disulfide is still able to expose its hydrophobic core in order to bind σ$^R$. This in turn explains why Cys11 is required for redox sensing. Finally, the structure of RsrA$^{ox}$ explains why the trigger disulfide is degenerate since the side chains of Cys41Ser and Cys44 are presented to Cys11 from consecutive turns of helix III$^N$ such that they can both form a disulfide bond (Fig. 7c). The similarity of the HSQC spectra of the two oxidized forms of RsrA (Cys11–Cys41 and Cys11–Cys44; Supplementary Fig. 6b) further suggest their structures are likely to be similar.

## Discussion

The mechanisms by which sensor proteins respond to oxidative stress in bacteria are varied but fall broadly into two groups, those that contain metal centres such as the transcriptional repressor PerR[32] and chaperone Hsp33 (ref. 33), and those that have reactive cysteines, such as the transcription factors OxyR[34] and OhrR[35]. Oxidation of tetrameric OxyR by hydrogen peroxide induces disulfide bond formation within OxyR monomers, the resulting conformational changes converting it from a transcriptional repressor into an activator[36]. The OhrR family of dimeric transcriptional repressors are derepressed by organic hydroperoxides following oxidation either of a single reactive

cysteine or through intersubunit disulfide bond formation[35,37]. The dimeric transcriptional repressor PerR contains two metal centres, a structural $Zn^{2+}$ site containing histidine residues[32], and a regulatory site that in its $Fe^{2+}$-bound state is responsive to oxidation. Derepression of PerR by hydrogen peroxide occurs through oxidation of metal-binding histidine residues following the generation of $HO\cdot$ by Fenton chemistry at the $Fe^{2+}$ site. Hsp33 is a heat-shock protein that becomes activated during oxidative stress. A single zinc ion is coordinated by four cysteines in the C-terminal domain of Hsp33. Following oxidation with $H_2O_2$, intramolecular disulfide bonds form between the zinc-ligating cysteines, expelling bound zinc and forming a dimeric chaperone[38]. Zinc is also expelled from $RsrA^{red}.Zn^{2+}$ on oxidation to $RsrA^{ox}$, but in this instance release of zinc accelerates the dissociation rate of its complex with $\sigma^R$ before formation of the trigger disulfide bond between its zinc-coordinating cysteine residues. Intriguingly, $RsrA^{red}.Zn^{2+}$ has the same redox potential as OxyR[39] even though the two proteins share no structural similarity and sense different oxidative stresses by completely different mechanisms. Finally, RsrA is the first example of an oxidative stress sensor that responds to oxidation by sequestering hydrophobic residues required to stabilize the protein–protein interaction with its cognate transcription factor back into its own hydrophobic core. The same residues are involved in stabilizing three distinct structural states of the anti-sigma factor, $RsrA^{red}.Zn^{2+}$, $RsrA^{red}.Zn^{2+}$–$\sigma^R$ complex and $RsrA^{ox}$ (Fig. 7).

$RsrA^{red}.Zn^{2+}$ and $RsrA^{red}.Zn^{2+}$–$\sigma^R$ complex are equally reactive towards oxidants however given the high affinity of the complex ($K_d \sim 0.7$ nM) and the co-expression of their genes in *S. coelicolor* it is likely that the complex is the redox sensor. The distance between the sulfur atoms of Cys11 and Cys44 in the metal-binding site is $\sim 3.5$ Å for both $RsrA^{red}.Zn^{2+}$ and the $RsrA^{red}.Zn^{2+}$–$\sigma^R$ complex. Formation of a disulfide bond between these sulfur atoms reduces the interatomic distance to 2.05 Å. This small ($\sim 1.5$ Å) change in bond length and valency precipitates a $\sim 20$ Å contraction of the $\sigma^R$-bound state to form $RsrA^{ox}$. Hence, oxidation of the RsrA thiols represents a highly efficient means of amplifying a small, local, chemical change into a large-scale structural collapse that blocks binding to $\sigma^R$.

Campbell *et al.*[13] have previously defined an ASD fold for both zinc-binding and non-zinc-binding anti-sigma factors. It is clear from the present work that this fold pertains only to the sigma-bound states of anti-sigma factors. The structure of $RsrA^{red}.Zn^{2+}$ illustrates a very different four-helix bundle structure when not bound to its cognate sigma factor. We term this form of the anti-sigma factor the ZAS fold (Supplementary Fig. 8). Structure-based sequence alignments of ZAS proteins recovered from bacterial genomes show that in addition to the zinc-binding ZAS motif these proteins all share conserved or conservatively substituted hydrophobic residues. Those displayed from helices I and II stabilize the ZAS and ASD folds of RsrA, while those displayed from helices III and IV stabilize the ZAS fold in the absence of sigma factor, but are then recruited to bind the sigma factor (Supplementary Fig. 8b).

A poorly understood feature of bacterial ZAS proteins is that they respond to diverse stresses. The present work shows why ZAS proteins with HHCC-type sequence motifs cannot be sensors of disulfide or peroxide stress, implying that additional domains or sequences are involved in sensing other types of stress. This is the case for the HHCC-type ZAS protein ChrR from *R. sphaeroides* where its cupin-like domain is the sensor of singlet oxygen stress. Sequence surveys of the ZAS sequence motif identifies >1,100 ZAS proteins distributed widely amongst bacterial phyla (Supplementary Fig. 8b,c). We found a statistically significant correlation of HHCC-type ZAS proteins with cupin-7

domains (66/498 HHCC-type ZAS sequences versus 1/665 CHCC-type ZAS sequences). How additional sensor domains disrupt the protein–protein interactions of ZAS protein–sigma factor complexes is unknown. We speculate that the modulation by zinc of the binding affinity of these complexes may be a route by which diverse signals could result in ZAS protein dissociation from the sigma factor. Zinc dissociation could be brought about by the destruction of zinc ligands by ROS, as occurs in the transcriptional repressor PerR, or proteolytic cleavage, as occurs in the ZAS protein RsiW[40].

## Methods

**Protein purification.** RsrA and its mutants were expressed and purified from *Escherichia coli* BL21(DE3) pLysS using a His-tag that was subsequently removed by thrombin cleavage. $RsrA^{ox}$ was generated by treating the protein with 0.1 mM diamide for 30 min. The reduced and oxidized forms of the proteins for structure determination were prepared on a Vydac C8 semi-preparative HPLC column; buffer A was 5% acetonitrile, 95% $H_2O$ and 0.1% trifluoroacetic acid, and buffer B was 95% acetonitrile, 5% $H_2O$ and 0.1% trifluoroacetic acid. Zinc reconstitution of reduced RsrA is described below. Uniformly $^{15}N$-labelled and $^{15}N,^{13}C$-labelled protein was prepared using minimal growth media supplemented with $^{15}N$ ammonium chloride and $^{13}C$ glucose (Cambridge Isotope Laboratories). $\sigma^R$ was purified without using purification tags. Residues 5–227 were subcloned into pET21a using NdeI and BamHI restriction sites with a C-terminal stop codon to enable tagless expression of $\sigma^R$. $\sigma^R$ in pET21a was expressed in *E. coli* BL21(DE3) cells at 37 °C in LB and expressed with 1 mM isopropyl-β-D-thiogalactoside for 3 h post induction. Cells were collected by centrifugation and lysed by sonication in 50 mM Tris (pH 7.5) and 100 mM NaCl. Lysate was purified by centrifugation and the supernatant was further purified by ammonium sulfate precipitation. $\sigma^R$ precipitated at 40% saturation ammonium sulfate. The pellet was resuspended in 50 mM Tris (pH 7.5) and 10 mM NaCl, and loaded onto a Q-Sepharose column. Purified $\sigma^R$ was concentrated and then loaded onto a Hi-Load S75 column, and eluted in 50 mM Tris (pH 7.5) and 100 mM NaCl. All $\sigma^R$ mutants were similarly purified. $RsrA^{red}.Zn^{2+}$–$\sigma^R$ complexes were generated by co-expressing the two genes in the same expressing strain of *E. coli*. *sigR* was cloned into pCDF-DUET1 vector between the BamHI and HindIII in MCS1. pET15b-*rsrA* and pCDF-DUET1-*sigR* were co expressed in pLysS-Rosetta2 cells. Protein was purified using the N-terminal histidine tag of RsrA on a $Ni^{2+}$-charged nitrilotriacetic acid (NTA) column and then subsequently purified by size-exclusion chromatography using a Superdex-75 column.

**Site-directed mutagenesis.** Mutations were made using the QuickChange lightning Multi Site-Directed mutagenesis kit (Agilent) following the manufacturer's instructions. Primer sets were purchased from MWG-Biotech (UK). Introduction of mutations was confirmed by DNA sequencing (Source Biosciences).

**Stoichiometry of zinc binding to RsrA.** Controversy surrounds the role of zinc in RsrA function. Li *et al.*[25] and Bae *et al.*[26] have suggested that zinc has little role to play in $\sigma^R$ binding and redox sensing, respectively, while Heo *et al.*[31] suggest that multiple zinc ions bind to RsrA to modulate its redox reactivity[31]. We re-assessed the stoichiometry of zinc binding before dissecting the redox-sensing mechanism. Using both wild-type RsrA and RsrA*, in which the four non-essential cysteines (Cys3, Cys31, Cys61 and Cys62) were mutated to alanine, native state nanoelectrospray mass spectrometry showed that both proteins in the reduced state bind multiple zinc ions (Supplementary Fig. 1a shows data for the wild-type protein). Zinc titrations using RsrA* in circular dichroism and mass spectrometry experiments indicated that only a single zinc ion is required to stabilize the RsrA fold (Supplementary Fig. 1a–d). Previous estimates of the affinity of RsrA for zinc documented an equilibrium dissociation constant ($K_d$) of $\sim 10^{-17}$ M (ref. 26), which is reasonable to assume pertains to this structural metal ion. We refer to this reduced, zinc-bound state of RsrA as $RsrA^{red}.Zn^{2+}$. This form of RsrA (dissolved in 50 mM Tris (pH 7.5), 100 mM NaCl and 1 mM dithiothreitol (DTT)) was obtained by treating all preparations of the protein with EDTA (1 mM, 15-min incubation), followed by desalting on a HiTrap column (5 ml) and, unless specified, stoichiometric zinc (Alfa Aesar, 99.9% purity) added back to the protein.

**Fluorescence spectroscopy.** Changes in the intrinsic tryptophan fluorescence emission of $\sigma^R$ on binding $RsrA^{red}.Zn^{2+}$ were monitored using a Horiba Fluor-oMax4 spectrofluorimeter in 50 mM Tris (pH 7.5) buffer containing 100 mM NaCl and 2 mM DTT at 25 °C. Samples were excited at 295 nm and emission spectra recorded from 310 to 450 nm. Experiments were conducted using 3 nm slit widths. $\sigma^R$ was kept at 1 µM and the concentration of RsrA varied from 0.1 to 2 µM from which the fractional change in fluorescence change was determined.

**Circular dichroism spectroscopy.** Apo-RsrA was prepared by incubating the protein with 1 mM EDTA for 15 min at 4 °C before buffer exchanging to the

experimental buffer using a HiTrap column. Samples were incubated in 50 mM Tris (pH 7.5) buffer containing 100 mM NaCl and 2 mM DTT before desalting in 10 mM Tris (pH 7.5) buffer using a HiTrap column. Experiments were conducted immediately on a Jasco-J815 CD spectrometer at 20 °C using a 1-mm path length cuvette. Data collected as ellipticity in millidegrees were converted to mean residue ellipticity, ($\theta$; degrees cm$^2$ per dmol residue).

**Native state mass spectrometry of protein complexes.** Intact mass spectrometry measurements were performed on a Waters Synapt G2 HDMS modified for high mass transmission (ref. 41). Samples were buffer exchanged into 25 mM ammonium acetate using BioSpin 6 (Bio-Rad) columns before mass spectrometry (MS) analysis. Typically, each measurement was performed by loading 3 μl of protein sample into gold-coated nanospray capillaries prepared in-house[42] and loaded into a static nanospray block providing the spray voltage. Electrospray was induced by applying a potential of between 0.9 and 1.2 kV to the capillary. A sample cone of 50 V was used to capture the charged droplets, with a source backing pressure of 5.5 mbar. The instrument was operated in time-of-flight mode (no ion mass spectrometry separation), with the trap and transfer collision cells held at an acceleration voltage of 10 and 5 V, respectively (with an argon pressure of $2 \times 10^{-2}$ mbar). Data were processed using the MassLynx software. For high-resolution MS measurement, a Thermo Scientific Q-Exactive hybrid quadrupole-Orbitrap mass spectrometer was used to measure the presence/absence of the disulfide in the RsrA* Cys11His–σ$^R$ complex. The instrument was modified for high mass measurement as described elsewhere[43,44] and optimized for retention of non-covalent interactions. Hardware alterations include a lower radio-frequency (RF) applied to the selection quadrupole and a gas line allowing higher pressures to be achieved in the higher-energy collisional dissociation (HCD) cell. The instrument was operated in 'native mode', where both the RF frequencies applied to transfer optics and Orbitrap voltages were optimized for high mass species. Ions were generated in the positive ion mode from a static nanospray source using gold-coated capillaries prepared in house, then passed through a temperature-controlled transfer tube (set to 20 °C, readback 31 °C), RF lens, injection flatapole and bent flatapole. After traversing the selection quadrupole, which was operated with a wide selection window (1,000–10,000 m/z), ions were trapped in the HCD cell before being transferred into the C-trap and Orbitrap mass analyser for detection. Transient times were 128 ms (35,000 at m/z = 200) and automatic gain control (AGC) target was $1 \times 10^6$ with a maximum fill time of 100 ms. No additional HCD or in-source activation was applied. Argon was used as the collision gas and the pressure in the HCD cell was maintained at around $1.3 \times 10^{-9}$ mbar. To achieve both representative mass measurement and good separation of the bound species, ~100 spectra, each of 25 microscans, were acquired then averaged using Thermo Scientific Xcalibur 2.1. Masses were calculated using the three most abundant charge states using an in-house software tool. Calibration was performed before mass measurement up to 11,300 m/z using clusters of CsI.

**Redox potential measurements.** The redox potential of the RsrA$^{red}$.Zn$^{2+}$–σ$^R$ complex was determined with reference to a GSH–GSSG (glutathione–glutathione disulfide) redox couple and monitoring the change in tryptophan fluorescence emission at 343 nm (excitation at 295 nm) of the complex (see above) and at 500 nm using the PAR assay in a spectrophotometer to determine the fraction of dissociated zinc (see below). Glutathione and glutathione disulfide were from Sigma-Aldrich. Fluorescence titration experiments were carried out in 50 mM Tris (pH 7.5) buffer and 100 mM NaCl at 25 °C. All buffers were degassed and purged with N$_2$ gas before the use. In all experiments, GSH was kept constant at 0.5 μM, while GSSG was varied from 100 μM to 100 mM. The RsrA$^{red}$.Zn$^{2+}$–σ$^R$ complex was at 2 μM. All samples were incubated with the redox couple for 2 h before data collection. Two hours was deemed as having reached equilibrium since longer incubations had no effect. Data were corrected for the inner-filter effect, as described[45]. Inclusion of the PAR reagent does not influence redox titration experiments, since its affinity for zinc is several orders of magnitude lower than that of RsrA[46]. Positive and negative controls were collected at each redox couple concentration. Positive controls had 1 equiv. Zn$^{2+}$ along with the PAR reagent and the GSH couple. Negative controls had the couple and PAR. The difference between the positive and the negative control gave the total signal change for that concentration of redox couple. Ratio of the experiment over the total signal change gave the ratio of zinc released. The fraction of RsrA$^{red}$.Zn$^{2+}$–σ$^R$ complex dissociated and the fraction of zinc released were used to determine the fraction of reduced RsrA (R). The variation of reduced RsrA against $\frac{[GSH]^2}{[GSSG]}$ was fitted to equation (1) to determine $K_{eq}$ in M units. $K_{eq}$ thus obtained was used in the Nernst equation (equation 2) to calculate RsrA's redox potential.

$$R = \frac{\frac{[GSH]^2}{[GSSG]}}{K_{eq} + \left(\frac{[GSH]^2}{[GSSG]}\right)}, \qquad (1)$$

where R is the reduced fraction of RsrA.

$$E_0^{RsrA} = E_0^{GSH/GSSG} - \left(\frac{RT}{2F}\right) \times \ln K_{eq}, \qquad (2)$$

where $E_0^{RsrA}$ is the redox potential of RsrA. $E_0^{GSH/GSSG}$ is the redox potential of glutathione, which is $-240$ mV at 25 °C, pH 7.5 (ref. 47). R is the gas constant 8.314 J K$^{-1}$ mol$^{-1}$ and F is the Faraday constant, 96.485 J mV$^{-1}$ mol$^{-1}$.

**ITC measurements.** All experiments were conducted using a Microcal ITC$_{200}$ instrument at 35 °C in 50 mM Tris (pH 7.5) buffer containing 100 mM NaCl and 2 mM DTT. No heats of binding could be detected at 25 °C. The affinity of the RsrA$^{red}$.Zn$^{2+}$–σ$^R$ complex was too high to be determined by direct titration and so was obtained by competition ITC. A weak binding mutant of RsrA, RsrA* Cys11Ser Cys44Ser ($K_d$, 185 nM), was included in the cell with σ$^R$ as a competitor. RsrA$^{red}$.Zn$^{2+}$ was titrated into this mixture. For weaker binding mutants of RsrA or in experiments where zinc was omitted, thermodynamic data were obtained by direct titration where 10 μM RsrA was in the cell and 100 μM σ$^R$ in the syringe. Where zinc was included in the experiment, RsrA was preincubated in 50 mM Tris (pH 7.5), 100 mM NaCl, 1 mM ZnCl$_2$ and then buffer exchanged on a 5-ml HiTrap column into 50 mM Tris (pH 7.5), 100 mM NaCl and 2 mM DTT. Zinc could not be included in the cell due to the precipitation of σ$^R$ during the titrations. All experiments were carried out immediately after buffer exchange and in triplicate. Averages and s.d.'s of the obtained parameters are reported from triplicate experiments. Data were analysed using the manufacturer's software assuming a single binding site model. Competition ITC titrations were performed at the same temperature and in the same buffer conditions by injecting 100 μM RsrA into 10 μM σ$^R$ containing 50 μM RsrA* Cys41Ser. Binding isotherms were analysed using the manufacturer's software for a competitive binding model[48].

**Association and dissociation kinetics of the RsrA–σ$^R$ complex.** *Association kinetics.* Stopped-flow fluorescence experiments were performed on an Applied Photophysics SX20MV instrument set-up for 1:1 single mixing and thermostated using a circulating water bath. An excitation wavelength of 295 nm was used for the excitation of σ$^R$'s two tryptophans (RsrA does not contain tryptophan), while a 320-nm filter was used to collected the fluorescence emission. The manual entrance and exit slits were set to 2 mm (bandpass = 4.65 nm/mm). Experiments were carried out at 25 or 35 °C in 50 mM Tris (pH 7.5) buffer containing 100 mM NaCl and 2 mM DTT. RsrA was preincubated in 50 mM Tris (pH 7.5), 100 mM NaCl, 10 mM DTT, 1 mM ZnCl$_2$ and buffer exchanged using a 5-ml HiTrap column into 50 mM Tris (pH 7.5), 100 mM NaCl and 2 mM DTT before use. All association experiments were carried out under pseudo-first-order conditions. The concentration of σ$^R$ was kept constant at 125 nM and RsrA ($\pm$ Zn$^{2+}$) varied from 1.25 to 2.5 μM in 250 nM increments. The kinetic traces were fitted to a single-exponential rate equation by nonlinear least square regression on the manufacturer's software. Values of $k_{obs}$ were then plotted against RsrA concentration to determine the bimolecular association rate constant ($k_{on}$). Data presented are averages of three traces in each stopped-flow experiment and each experiment was performed three times. Quoted errors are the s.d. from the three repeats. Apo-RsrA was prepared by incubating the protein with 1 mM EDTA for 15 min at 4 °C before buffer exchanging to the experimental buffer on a HiTrap column.

*Dissociation kinetics.* Competition experiments were conducted to determine the dissociation rate constant of the RsrA–σ$^R$ complex by mixing 2.5 μM RsrA$^{red}$.Zn$^{2+}$–σ$^R$ complex with 25 μM σ$^R$ Trp88Ile Trp119Ile. σ$^R$ Trp88Ile Trp119Ile binds RsrA$^{red}$.Zn$^{2+}$ with the same affinity as wild-type σ$^R$ (Supplementary Fig. 3a), but does not produce a change in tryptophan fluorescence. All experimental conditions were as described above. The dissociation trace was fitted to a single-exponential rate to determine $k_{off}$. All experiments were repeated three times, and averages and s.d.'s were reported. Apo-RsrA was prepared as described above. Zinc-bound RsrA was prepared by incubating either 1 or 3 equiv. of Zn$^{2+}$ for 15 min at 4 °C before buffer exchanging to the experimental buffer on a HiTrap column.

**Oxidation kinetics of the RsrA–σ$^R$ complex.** All experiments were conducted at 25 °C in 50 mM Tris (pH 7.5) buffer containing 100 mM NaCl. Buffers were thoroughly degassed and purged with N$_2$ gas before use.

*Oxidation by diamide.* The reaction of diamide with thiols can be followed spectrophotometrically at 320 nm. Each diamide molecule oxidizes two thiols forming a disulfide and in the process diamide is converted from its diazene to hydrazine form (Supplementary Fig. 2). Only the diazene form absorbs at 320 nm (ref. 30). For second-order experiments, diamide (25 μM) was added to RsrA$^{red}$.Zn$^{2+}$–σ$^R$ complex (25 μM), prepared by incubating EDTA-treated complex with 1 equiv. Zn$^{2+}$. Oxidation of the complex ($\pm$ Zn$^{2+}$) was followed by the change in absorbance at 320 nm for 1,200 s using an Applied Photophysics SX20MV stopped-flow apparatus. The concentration of diamide at various time points was computed by determining the ratio of diamide consumed relative to a control in which 25 μM diamide was converted to the hydrazine form using 500 μM DTT. Raw data were then linearized by plotting the variation of 1/[diamide] against time from which the second-order rate constant for diamide-induced oxidation was determined. Zinc release from the RsrA$^{red}$.Zn$^{2+}$–σ$^R$ complex on oxidation with diamide was measured using 100 μM PAR, which absorbs at 500 nm on zinc binding[46]. The second-order rate of diamide-induced release of Zn$^{2+}$ was determined as described above using the PAR assay. For pseudo-first-order experiments, only zinc release was monitored from the RsrA$^{red}$.Zn$^{2+}$–σ$^R$ complex (2 μM), where diamide was varied from 25 μM to 20 mM. Traces were fitted to a single-exponential rate equation to obtain the observed Zn$^{2+}$ release rate, $k_{obs}$. Plots of $k_{obs}$ versus diamide concentration were hyperbolic, the data fitted to the Michealis–Menten equation from which values for

$K_1$ and $k_2$ were extracted (Supplementary Fig. 2b). The average and s.d.'s reported are from triplicate experiments.

*Oxidation by $H_2O_2$.* Buffer and experimental conditions were as described above. Only pseudo-first-order experiments were conducted with $H_2O_2$. Zinc release was monitored in a stopped-flow apparatus using the PAR assay where the RsrA$^{red}$.Zn$^{2+}$–σ$^R$ complex (2 μM) was incubated with varying concentrations of $H_2O_2$ (1–10 mM). PAR was kept at 100 μM. Observed rates of Zn$^{2+}$ release were plotted against the concentration of $H_2O_2$ from which the bimolecular rate constant was obtained as above. The change in intrinsic tryptophan fluorescence of σ$^R$ on complex formation was also exploited to follow the rate of complex dissociation on treatment with $H_2O_2$. RsrA$^{red}$.Zn$^{2+}$–σ$^R$ complex (2 μM) was incubated with varying concentrations of $H_2O_2$ (1–10 mM) and the extent of complex dissociation determined by fluorescence spectroscopy (excitation wavelength, 295 nm and emission wavelength, 343 nm) using a Fluoromax4 spectrometer and 10-mm path length quartz cuvettes. The average and s.d.'s reported are from triplicate experiments.

**Crosslinking-based homology modelling of the RsrA$^{red}$.Zn$^{2+}$–σ$^R$ complex.**
Crosslinking was used to constrain homology models of the complex. A concentration of 5 μM purified RsrA$^{red}$.Zn$^{2+}$–σ$^R$ complex in 20 mM Hepes (pH 7.5) buffer containing 50 mM NaCl and 2 mM DTT was incubated with 2 mM of 1:1 mixture of BS2G-$d_0$ and BS2G-$d_4$ (bis(sulfosuccinimidyl) 2,2,4,4-glutarate) or 1:1 mixture of BS3-$d_0$ and BS3-$d_4$. Crosslinking reactions were allowed to take place at room temperature for 30 min, and then quenched with Tris–HCl (pH 7.5) to a final concentration of 100 mM. Samples were then separated on 15 % SDS–polyacrylamide gel electrophoresis and crosslinked bands excised, reduced with 10 mM DTT at 56 °C for 30 min, and alkylated with 50 mM iodoacetamide (Sigma-Aldrich) in the dark at room temperature for 20 min. A measure of 10 ng μl$^{-1}$ trypsin (Porcine, Promega) was then added to cover gel pieces and digestion was allowed to continue overnight at 37 °C while shaking. Tryptic peptides were separated on an EASY-nLC[49] 1000 UHPLC system (Proxeon) and electrosprayed directly into a Q-Exactive mass spectrometer (Thermo Fischer Scientific) through an EASY-Spray nano-electrospray ion source (Thermo Fischer Scientific). Peptides were trapped on an in-house packed guard column (75 μm inner diameter × 20 mm, reprosil C18, 3 μm, 120 Å) using solvent A (0.1% formic acid in water) at a pressure of 500 bar and then fractionated using an EASY-spray Acclaim PepMap analytical column (75 μm inner diameter × 500 mm, RSLC C18, 2 μm, 100 Å) eluted with a linear gradient (7–31% solvent B (0.1% formic acid in acetonitrile in 30 min) at a flow rate of 200 nl min$^{-1}$. Full-scan MS spectra were acquired in the Orbitrap (scan range 350–2,000 $m/z$, resolution 70,000, AGC target 3e6, maximum injection time 100 ms). After the MS scans, the 10 most intense peaks were selected for HCD fragmentation at 30% of normalized collision energy. HCD spectra were also acquired in the Orbitrap (resolution 17,500, AGC target 5e4, maximum injection time 120 ms) with first fixed mass at 100 $m/z$. Charge states 1+ and 2+ were excluded from HCD fragmentation.

MS data were converted into mgf format using MSconvert from the ProteoWizard toolbox[50] and searched using the pLink software[51]. The database contained the target proteins only (RsrA and σ$^R$). Search parameters were as follows: maximum number of missed cleavages = 2, fixed modification = carbamidomethyl-Cys, variable modification 1 = oxidation-Met, variable modification 2 = Glu to pyro-Glu, mass shift of BS2G-$d_0$ = 96.02113, mass shift of BS2G-$d_4$ = 100.04583, mass shift of BS3-$d_0$ = 138.06809, mass shift of BS3-$d_4$ = 142.09279, mass accuracy filter = 20 p.p.m. for precursor ions with consideration of the first five isotopic peaks, MS2 tolerance = 20 p.p.m. Data were initially filtered by E-value ($E < 1.98e - 4$). Crosslinks were further inspected by checking the presence of the peak pair in the MS spectra generated by the $d_4/d_0$ mixture, as well as fragmentation patterns.

Three initial atomic models of RsrA–σ$^R$ complex were built based on the crystal structures of RseA–σ$^E$ (ref. 52), ChrR–RpoE[13] and RslA–σ$^L$ (ref. 22) using MODELLER V9.12 (ref. 53). Structure-based alignment of the above three models with the NMR structure of RsrA$^{red}$.Zn$^{2+}$ was used to define the secondary structure of RsrA in the complex. Each model was checked against the experimental crosslinks. A model based on RseA–σ$^E$ alone did not satisfy the crosslinks observed and hence discounted while the pattern of crosslinks discounted the conformational change seen for σ$^4$ in the ChrR–RpoE complex[13]. A final composite model was built for σ$^R$ using the crystal structures of the σ$^2$ domain from σ$^R$ (ref. 54) and σ$^4$ domain from the RslA–σ$^L$ (ref. 22) complex, and for RsrA$^{red}$.Zn$^{2+}$ from ChrR and RslA from their complexes with σ$^E$/RpoE and σ$^L$, respectively. Lysine residues from helix IV of RsrA gave crosslinks to different regions of σ$^R$ consistent with previous observations that it can exist in different conformations in ZAS protein complex structures, contacting either the σ$^2$ or σ$^4$ domains[13]. One hundred models were generated and the lowest-energy model (assessed by discrete optimized protein energy (DOPE) energy function) selected for further analysis.

**NMR Spectroscopy.** RsrA* Cys41Ser concentrations in all NMR samples were 1–1.5 mM. Oxidized samples were prepared in 20 mM Tris–HCl (pH 7.1) and 95% $H_2O$/5% $D_2O$. For reduced state samples, spectra were acquired in 20 mM Tris–HCl (pH 7.1), 5 mM DTT, 2 mM ZnCl$_2$ and 95% $H_2O$/5% $D_2O$. Where necessary (for example, for acquisition of homonuclear two-dimensional nuclear Överhauser

enhancement spectroscopy (NOESY) and total correlation spectroscopy (TOCSY) experiments), deuterated Tris and DTT (Cambridge Isotope Laboratories) were used. With the exception of experiments to measure residual dipolar couplings (RDCs; see below), all the NMR experiments were performed on a Bruker Ultrashield 700 MHz spectrometer with triple ($^1$H, $^{15}$N and $^{13}$C) nucleus (TXI, Bruker Biospin) probe equipped with $z$ gradient coils, running TopSpin (Bruker BioSpin) software and belonging to the University of York Centre for Magnetic Resonance. All experiments were performed at 298 K with NOESY mixing times of 100–150 ms and a TOCSY mixing time of 50 ms.

*Processing and assignment of NMR spectra.* All spectra were processed using NMRPipe[55]. Backbone and side-chain assignments were made using a standard suite of three-dimensional (3D) triple resonance experiments HNCA, CBCANH and CBCA(CO)NH, HNCO and HN(CA)CO in addition to 3D $^{15}$N-$^1$H-HSQC-NOESY and $^{15}$N-$^1$H-HSQC-TOCSY. NOEs were assigned using 3D NOESY and two-dimensional $^1$H-$^1$H NOESY experiments. Assignments were made using CcpNmr Analysis version 1.0 (ref. 56). Chemical shift assignments were deposited at BMRB (accession numbers 25,955 and 25,956 for RsrA$^{red}$.Zn$^{2+}$ and RsrA$^{Ox}$, respectively).

*Residual dipolar couplings.* The $^1$H-$^{15}$N RDCs[57] were recorded using antiphase/in-phase experiments[58] at 14 T ($^1$H larmor frequency of 600 MHz) at the University of Southampton using a 5% C12E6/hexanol ($r = 0.64$) liquid crystalline medium as an alignment medium[59]. Data in the isotropic and aligned state for both the RsrA$^{red}$.Zn and RsrA$^{ox}$ samples were acquired at 298 K with sweep widths of 10,000 and 1,800 Hz, and acquisition times of 51 and 100 ms, for $^1$H and $^{15}$N, respectively. The observed deuterium splitting in the aligned state was 24 Hz for both the zinc-bound and oxidized samples. The data were processed with NMRPipe[55] and were further analysed with Sparky[60] using the built-in peak fitting module to determine the peak positions. Errors were estimated to be 1 Hz. The RDCs (40 restraints for RsrA$^{ox}$ and 29 for RsrA$^{red}$.Zn$^{2+}$) were analysed using Module[61] and incorporated into the structure calculations during refinement.

*Structure calculation and validation.* One hundred structures were calculated using the program CNS[62] (Table 1). The Ser42–Pro43 peptide bond was modelled in the *cis*-conformation (using the CIPP patch) in the structure of RsrA$^{red}$.Zn, based on the presence of an NOE between the Hα atoms of Ser42 and Pro43 and between Hβ of Ser42 and Hα of Pro43 and on the lack of NOEs between Ser42 Hα and Pro43 Hδ that would be characteristic of a *trans*-conformation. In RsrA$^{ox}$, however, this bond was modelled in the *trans*-conformation based on observed NOEs between Ser42 Hα and Pro43 Hδ. An additional patch was generated for inclusion of the zinc atom and modification of the zinc ligands (Cys11, His37, Cys41Ser and Cys44) in RsrA$^{red}$.Zn. Estimates of backbone dihedral angles were obtained using TALOS[63]; only restraints for residues that had 'good' predictions were included. Accessible molecular surface areas on a per residue basis (averaged over the 10 lowest-energy structures for RsrA$^{red}$.Zn$^{2+}$ and RsrA$^{ox}$) were calculated using WHAT IF[64]. Structure coordinates of RsrA$^{ox}$ and RsrA$^{red}$.Zn$^{2+}$ have been deposited in the Protein Data Bank (5frh and 5frf, respectively).

**Data availability.** Structure coordinates of RsrA$^{ox}$ and RsrA$^{red}$.Zn$^{2+}$ have been deposited in the Protein Data Bank (5frh and 5frf, respectively). NMR assignment data have been deposited at BMRB (accession numbers 25955 and 25956 for RsrA$^{red}$.Zn$^{2+}$ and RsrA$^{Ox}$ respectively). All other relevant data are available from the authors on request.

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

## Acknowledgements

We thank Nick Housden and Grigorios Papadakos (Department of Biochemistry, Oxford) for guidance on the kinetic and thermodynamic measurements in this paper. We are indebted to Mark Buttner and Morgan Feeney (John Innes Centre, Norwich) for comments on the manuscript, Joseph Gault (Department of Chemistry, Oxford) for help with acquiring high-resolution MS data and David Staunton (Molecular Biophysics Suite, Oxford) for help with biophysical measurements. This work was funded by BBSRC grant BB/I008691/1. J.R.P. holds a British Heart Foundation Senior Basic Science Fellowship (FS/12/36/29588), J.M.W. and L.P. would like to thank the Wellcome Trust for support of the Southampton NMR centre (grant no.: 090658/Z/09/Z) and the MRC for support of the Biomedical NMR Centre at Mill Hill (grant no.: U117533887).

## Author contributions

K.V.R. conducted the kinetic, thermodynamic and oxidation experiments, modelled the complex and prepared mutants and crosslinked samples. K.Z., L.P., J.M.W. and J.R.P.

determined NMR structures of RsrA. J.Y., J.T.S.H., S.M. and C.V.R. conducted the mass spectrometry experiments and associated analysis. M.-L.R.F. contributed to kinetic analysis and with C.Se. prepared the samples. C.Sh. and K.V.R. conducted the bioinformatics analysis. C.K. was the principal investigator and, along with K.V.R., prepared the manuscript.

## Additional information

**Competing financial interests:** The authors declare no conflict of interests.

