## [Peer Review File · Nature Communications]

Reviewer #1 (Remarks to the Author):

This study provides major new structural and mechanistic insights into RsrA, which is a member of a class of regulatory proteins, zinc-containing anti-sigma (ZAS) factors, that are important and widespread in bacteria including many pathogens. The work includes the first structure of a ZAS protein in the absence of the cognate sigma factor. The challenging set of experiments appear to be very well executed and presented and the findings are likely to be of widespread interest in part due to the proposed striking reorganisation of the hydrophobic core on binding the sigma factor. One potential concern is the fact that the remodelling upon binding to SigR is based on a homology model. In my opinion these concerns are allayed by the cross-linking data and by mutagenesis data that confirm the importance of proposed hydrophobic interaction residues on SigR binding. I have only minor comments.

1. The redox couple GSH/GSSG was used to determine the redox potential of RsrA. The authors should make it clear that the reaction has gone to equilibrium, particularly since GSH/GSSG is not present in *Streptomyces*. Furthermore, have the authors accounted for any potential influence of PAR on the redox potential - since it competes with RsrA for zinc.
2. Data show clearly the importance of bound zinc in promoting formation of the SigR-RsrA complex. This is consistent with published work, which shows that zinc limitation induces SigR activity in vivo (doi: 10.1128/JB.01901-06)
3. Supplementary Fig 1 appears to be mislabelled.
4. p. 8 (Fig. 5) - I couldn't see the data for the C41S or C44S mutants - is this data not shown?
5. p. The conclusion that "in order for a ZAS protein to oxidatively dissociate from its target sigma factor the N-terminal zinc-coordinating residue within the ZAS motif must be a cysteine (CHCC)" is rather too general and appears to rule out the possibility that other members of this large family might have a different mechanism that does not involve this cysteine.
6. In Fig. 6 it appears that the oxidised figure is larger in size than the reduced.
7. p. 10 It is stated that the cross-linking data suggest that helix IV does not adopt a single conformation, but can contact both sigma 2 and sigma 4 domains. This seems ambiguous - isn't it the case that the data is consistent with either model such that helix IV would contact either domain 2 or 4.

Reviewer #2 (Remarks to the Author):

This study investigates the redox sensitive ZAS anti sigma factor RsrA of *S. coelicolor* that senses disulfide stress by formation of either C11-C41 or C11-C44 disulfides resulting in Zn-release and freeing the cognate SigR. The impact of a hydrophobic core that is utilized by RsrA to bind the sigma factor SigR is investigated at the structural level and Zn is important to maintain this complex. The authors show that oxidation of RsrA is limited by the Zn-release. They show the structural rearrangement of RsrA upon oxidation by formation of the trigger disulfide of C11-C44 using the C41Ser mutant where the SigR-binding residues are sequestered in its hydrophobic core preventing interaction with SigR.

In general this is a very interesting paper, since thus far the structure of the RsrA-SigR complex has not been resolved and the identification of the RsrA hydrophobic core which is important for the RsrA interaction with SigR are novel findings of this study. Furthermore, the structures of reduced and oxidized RsrA are not yet known in comparison and not how the complex with SigR is affected and hence this study shows first time the dramatic changes in the structure of RsrA upon formation of the disulfide and the changes in the accessibility of the hydrophobic core.

The methods are conclusive for all in vitro results and the data are solid and detailed explained. However, I have major concerns concerning the various investigated RsrA Cys-Ser and Cys-His mutants that have been extensively characterized for its Zn-binding ability and its effect on the oxidatively induced dissociation of the RsrA-Zn SigR complex in vitro.

Major comments:

1) My basic concern is the trigger C11-C44 disulfide that was formed in vitro in the oxidized structure of the C41Ser mutant was not confirmed using the the wild type protein. Furthermore, there are no supporting in vivo data shown which of the various Cys-Ser and Cys-His mutants analyzed in this study are functional and responsive in vivo. The wild type protein could form also the C41-C44 disulfide which was shown in the mass spec data for the C11His mutant (Fig 5c) that was also shown to bind Zn. The authors demonstrate that this C11His mutant did not result in oxidation-induced dissociation of the RsrA-SigR complex. However, it has been not investigated how the response of the RsrA C11His mutant is in vivo to oxidative stress. Furthermore, there are also RseA homologs in *C. glutamicum* and *Mycobacteria* that do not have the N-terminal Cys and RseA also responds to oxidative stress and controls in *Cglutamicum* a sub-regulon of the RshA regulon. Thus, there are certainly ZAS proteins that sense oxidative stress without this N-terminal Cys among the Actinomycetes. I think the important question is whether the trigger disulfide is really C11-C41/ C11-C44 or if this could be also the C41-C44 disulfide as in other Zn-based redox switches where disulfide form in CxxC Zn-binding motifs (e.g. Hsp33) ? Thus, I think concerning all these Cys-His-mutants, there must be detailed in vivo results shown to confirm if the RsrA Cys variants are functional in vivo to bind SigR and if these are responsive to diamide and H₂O₂. I assume that the formation of the C41-44 disulfide could have different effects on the structural re-arrangements and should be investigated in more detail also on the structural level.

Please add Northern blot results to show the transcription of selected SigR-dependent target genes in all of these single and multiple analyzed Cys-His mutants to analyze which are functional and responsive in vivo and which Cys is important for redox-sensing. How does the C11Ser and C11His variants behave to diamide stress in vivo ?

2) For the mass spec data (Fig. 5) please add also the results about oxidation of the wild type RsrA protein and detailed fragment ion spectra, mass deviations and scores should support the oxidation of the specific Cys-residues of the Cys peptides. However, the C41-C44 disulfide could also form in vitro artificially during the mass spectrometry analysis and hence MS-based thiol-trapping assays (e.g. OxICAT analysis) are required of the wild type protein to identify the specific Cys residues involved in disulfide formation in vivo. To determine the in vivo-relevance of the trigger disulfide, the OxICAT analysis should be performed also for the wild type RsrA protein under reduced and oxidized conditions after pull-down of the RsrA protein from cell extracts. I think concerning different expected structural rearrangements for C41-C44 and C11-C41 disulfides, it is important to define the nature of the disulfide formed in RsrA in vivo.

3) Figure 6 shows the solution structures of RsrA_{red} and RsrA_{ox}, but the reduced structure was obtained from the C41Ser mutant. How is the structural change in this hydrophobic SigR-binding core in the RsrA wild type protein ? I think it would be important to get insights into the structural changes of the wild type protein and the nature of the disulfide bond.

Specific comments:

- 1) In the title and abstract it should be added that RsrA is the anti-sigma factor from *Streptomyces coelicolor* which is not clear here.
- 2) In all figures, the number of replicates, error bars and P-values are required. There are no statistics in Figure 2-5.
- 3) Figure 2: Please add the molar ratio of SigR and RsrA in the legend used for the complex formation and Trp fluorescence changes.
- 4) Figure 4: The changes in the Trp fluorescence should be also included for the dissociation of the RsrA:Zn complex by diamide stress.
- 5) Figure 5cd: Present detailed fragment ion spectra for the Cys peptides as outlined above and % oxidation of each Cys using MS-based thiol-trapping for the RsrA wild type protein reduced and oxidized (in vitro and in vivo).
- 6) Figure 6c: Structure of RsrA_{ox} should be shown for the wild type protein (see above).

Reviewer #3 (Remarks to the Author):

I found the paper to be very interesting and compelling. It is rare to be able to capture a system in such diverse states while backed up with a variety of spectroscopic and other biophysical techniques. Overall I think the paper is quite publishable.

I personally would have found it helpful to have a kinetic scheme as a figure, with the various zinc bound and unbound states, and the sigma bound or unbound.

The scheme could be complete with estimated on and off rates where known, thus summarizing all the results neatly. The thermodynamic (equilibrium constant) connections between say zinc unbinding and sigma affinity would be able to be seen as natural consequences of a thermodynamic cycle.

The only other point I have is that the NMR should also be deposited at BMRB in addition to the PDB deposition of coordinates. This would be consistent with current standards for data sharing and reproducibility.

Response to referee comments for manuscript NCOMMS-16-01679-T

Referee #1

The referee provides minor comments.

1. The referee raises a reasonable point about possible competition of the zinc-chelating agent PAR with RsrA for zinc. However, since the affinity of PAR for zinc is several orders of magnitude weaker than that of RsrA for zinc this does not influence the redox titration measurements. This is further supported by the fact that there is good agreement between the PAR data and the fluorescence measurements that monitor the status of the RsrA- σ^R complex. We have added a sentence addressing this point in the Supplementary (p20) and added a relevant reference (Hunt et al 1985 Anal Biochem). In the same section we also emphasize that longer incubations in the redox potential measurements had no effect on the data and hence were deemed to have reached equilibrium.
2. This is also a good point. We have now added this reference (Owen et al 2007 J Bacteriol) to the main text along with an explanatory sentence (p6).
3. We thank the referee for noticing we had mislabeled the panels of Supplementary Figure 1! The figure legend has now been corrected to match the panels.
4. We did not add these data to figure 5 as they are identical to the WT data (mentioned in the figure legend). We feel the figure becomes too busy especially with the inclusion of zinc coordination cartoons to illustrate each construct.
5. We have modified the text to leave open this possibility, as suggested (p8, last sentence).
6. We have re-sized the panels.
7. We feel the suggested changes to the text actually make this more ambiguous! We have therefore left the text unaltered. The data are consistent with helix IV being dynamic and hence able to contact both helix 2 and helix 4.

Referee #2

The main thrust of this referee's comments is that we re-do all the work *in vivo* and solve new structures of wild type RsrA. There are three reasons why we believe this additional work is unwarranted and/or impractical. First, our paper is an *in vitro* mechanistic dissection of the RsrA- σ^R redox sensing mechanism, which the referee acknowledges ("The methods are conclusive for all (the) *in vitro* results...") as do the other two referees. The paper is already brimming with structural, biochemical and biophysical data, all necessary to describe the redox sensing mechanism of RsrA. A complete *in vivo* analysis, without clear justification, is we believe beyond the scope of the present work. Second, many of the *in vivo* experiments asked for by the referee have already been published (see below) and indeed are the foundation of the present study. Third, the structures we solved of mutant forms of RsrA are representations of this ZAS protein in its different functional states; it is difficult to envisage how one would capture these functional states if not by using mutants, a point touched on by referee 3. As referee 2 nevertheless acknowledges "the structures of reduced and oxidized RsrA are not yet known...and hence this study shows for the first time the dramatic changes in the structure of RsrA upon formation of the disulphide bond.." As we point out in the manuscript, there are currently no structures in the PDB for any ZAS protein in the absence of its sigma factor or following stress-induced inactivation, which is for a good reason - they are difficult to obtain!

Below we try to address the points raised by this referee:

1. The referee is concerned about the lack of evidence for the C11-C44 trigger disulphide in wild-type RsrA. As we point out in the introduction (p4, last paragraph), the degenerate nature of the trigger disulphide in RsrA, C11-C44 or C11-C41, has long been established; identified both by my lab (ref. 25) and that of Jung-Hye Roe (ref 26). The results from these early experiments, both of which used wild-type RsrA, are perfectly consistent with the *in vitro* data reported in the present paper. The referee appears to have missed this important point.

The referee asks for supporting *in vivo* data on the Cys/Ser and Cys/His mutations used in the paper. *In vivo* phenotypes of Ser mutations for all seven RsrA Cys residues have been reported previously (ref. 23). This work was mentioned in the introduction (p4, last paragraph). The referee does not make clear why repeating these experiments with His mutations, which were important in the context of the *in vitro* dissection of the redox sensing mechanism, would be informative.

The referee asks for *in vivo* data confirming C11H does not respond to oxidative stress. We have not included such data because in all cases where oxidative stress has been shown to inactivate a ZAS protein in actinobacteria *in vivo* (e.g. RshA in *M. tuberculosis*; Song et al 2003 *Mol Microbiol.* **50**, 949) the cysteine equivalent to C11 in RsrA is conserved (CHCC type motif). See Supplementary Figure 7c for an alignment of such sequences. The referee mistakenly claims there are actinobacterial ZAS proteins that are known to respond to oxidative stress yet lack the equivalent of C11 in RsrA. In doing so s/he points to work on RseA in *M. tuberculosis* and *C. glutamicum* (we could find no published work on *C. glutamicum*). First, RseA is a paralogue not an orthologue of *S. coelicolor* RsrA. Second, it responds to cell envelope stress (not oxidative stress) and its mechanism of inactivation is proteolytic release from its sigma factor (not oxidative dissociation) in a phosphorylation-dependent manner (e.g. Barik et al 2010 *Mol Microbiol.* **75**, 592). In other words, the physiological role of *M. tuberculosis* RseA- σ^E is unrelated to that of RsrA- σ^R and its mechanism of release is similarly unrelated to that of RsrA- σ^R thereby supporting (not contradicting) our contention that HHCC-type ZAS proteins are unlikely to be disulphide/peroxide stress sensors.

The referee maintains that a C41-C44 disulfide bond could be the trigger for RsrA oxidation. As explained above, previous work has shown that this is not the case. Moreover, our Cys-to-His mutational analysis coupled with the structures we have solved not only show that such a disulphide does not cause oxidative dissociation of the complex, but also provides the structural reasons why. We have added some text on p13 further detailing the structural reasons why a C41-C44 disulfide in RsrA does not cause oxidative dissociation from σ^R .

The referee requests Northern blots of strains carrying all the mutant RsrA proteins we have generated in our study to determine which are redox active *in vivo*. Paget et al. (ref. 23) did exactly this (but using S1 mapping rather than Northern blots) for all seven Cys-to-Ser mutations, which identified the three redox sensing cysteines in RsrA (Cys11, Cys41 and Cys44). All three are required because of the redundancy in trigger disulphide formation and because all three are involved in coordinating zinc. We see little point in repeating such experiments.

2. The referee asks for mass spec data of the oxidation products of the wild type protein. As outlined above and in the manuscript, these data have been reported previously. The referee also asks for *in vivo* trapping data to verify the trigger disulphide as C11-C44. Such experiments are beyond the remit of the present *in vitro* study and, we suggest, unnecessary.

3. The referee asks we define the structural changes in RsrA's hydrophobic core for the wild type protein binding σ^R since we rely on using the RsrA* C41S mutation to determine the structure of RsrA^{red}.Zn²⁺. As we explain in the manuscript, this has not been possible due to the lack of diffracting crystals for crystallography and poor spectral resolution in NMR spectroscopy of the wild type RsrA- σ^R complex. RsrA* C41S was used to obtain the structure of the reduced form of RsrA because this mutant gave good NMR spectra. The mutant binds both zinc and σ^R and forms the main trigger disulphide at equilibrium. Lastly, the cross-linking based homology model of the RsrA- σ^R complex is that of the wild-type complex, which we further validated by mutagenesis of hydrophobic residues (Table 1), again in a wild-type cysteine background.

Further minor points include:

The other referees did not raise the issue of the title so we have left it unchanged.

The referee asks for more statistical analysis for Figures 2-5. The quoting of P-values for such biochemical experiments is inappropriate. Many of the panels in fact do show error bars but these are too small to see. In some instances however error bars are not warranted. For example, pre-steady state data (Fig. 3 c and d). These are averages of multiple injections fitted independently and then replicate errors ($n = 3$) calculated and cited in the figure legend. Similarly, in Fig. 4, error bars are not appropriate for the main panels. These are kinetic traces, fitted as described in Materials and Methods and the average of three independent measurements given in the legend. The inset panels in this figure all have error bars, again too small to see. These have now been highlighted. Error analysis has been added to Supplementary Materials & Methods where this was absent.

We have not added molar ratios as these are evident from the inset in Figure 2.

As we point out in the manuscript (p7, second paragraph), we could not follow diamide-induced dissociation of the RsrA- σ^R complex by fluorescence spectroscopy because diamide absorbs at the wavelength used. Our use of hydrogen peroxide in these experiments was precisely so we could follow complex dissociation and zinc release.

The referee repeats his/her request for MS fragment ion analysis for Cys peptides of RsrA along with a request for “% oxidation of each cysteine’...’in vitro and in vivo”. As we have argued above, we see no need to repeat experiments reported over a decade ago and certainly when no rationale is given as to why such measurements are needed in the context of the present story. The degenerate nature of the RsrA trigger disulphide (Cys11-Cys41 or Cys11-Cys44) is well-established in the literature (refs. 25 and 26), and consistent with our NMR (Supplementary Figure 6) and Cys-to-His mutational data (Figure 5). Moreover, the NMR data show for the first time that the Cys11-Cys44 disulfide is the most stable of the two possible trigger disulfides at equilibrium.

The referee asks for the structure of wild type RsrA^{ox} but does not explain why (or how) this should be done. We have solved the structure of the most stable trigger disulphide form of RsrA at equilibrium (Supplementary Figure 6) while simultaneously blocking the formation of any mixed disulfides by removing Cys41. It is not clear why determining the same structure for the same protein but containing all seven cysteines would be any more informative. Indeed, it is unlikely this would even be possible given the likelihood of mixed disulfides making this a major technical challenge.

Referee #3

The referee suggests only minor changes.

1. A new kinetic scheme has been added as suggested. This has necessitated the addition of a new supplementary panel (Supplementary Figure 3d) along with associated text in the legend.
2. The NMR data have been deposited at BMRB as suggested by the referee: reference numbers 25955 and 25956 for zinc-bound and oxidized, respectively. (p27)

Reviewer #1 (Remarks to the Author):

I am happy that the authors have addressed all of my comments in my original review. I also feel that they have satisfactorily addressed the comments raised by the other two reviewers.

Reviewer #2 (Remarks to the Author):

I acknowledge the response to my comments and the difficulty of the structural analysis of the wild type RsrA protein. I also know about the previous in vivo analysis of all the Cys-Ser mutants.

But in that paper Cys-His mutants have been used that were not analyzed for the in vivo response in previous studies. Thus, I requested these results since I still think that in vivo results should supplement the in vitro biochemistry.

Secondly, as for SigE of *Cglutamicum* and SigE of *Mtuberculosis* (which are both each homolog to SigH of *Mtb* and *Cglut*), these are controlled by their cognate anti sigma RseA and our recent unpublished data clearly show that RseA is oxidized in vivo and the SigE regulon is induced under oxidative stress. The SigE regulon is even a subregulon of the SigH regulon in *Cglutamicum* and *Mycobacteria* and the promoter consensus sequence of both SigE and SigH is similar. Moreover, we have shown in vivo that indeed Cys41-44 are forming the disulfide in RseA of *Mycobacteria* under ROS stress in vivo using mass spec based thiol-trappings (OxICAT). Thus, I raised this point since we have the in vivo oxidation data for a quite close homolog of RshA, which is RseA in which Cys11 is absent.

We are writing this large-scale redox-proteomics and transcriptome paper up at the moment which perhaps should give further motivation in future work to analyze crystal structures of oxidized RseA in comparison to the reduced form which might be different compared to RsrA of *Scoelicolor*.

I accept the current study now as a biochemical work and follow up of previous studies about RsrA oxidation.

Reviewer #3 (Remarks to the Author):

The authors have responded to my criticisms in an appropriate way.

Response to referee comments for manuscript NCOMMS-16-01679-T (6 May 2016)

Referees 1 and 3 agree that we have satisfied all the comments of the three referees. Referee 2 now acknowledges that our paper is, as we had argued, a biochemical dissection of the RsrA- σ^R redox sensing mechanism. The referee makes no recommended changes other than to make the point they made in the last review concerning *in vivo* analysis of Cys-to-His mutations, again without a clear justification as to why these would strengthen the paper. Thereafter, the referee describes a series of unpublished data they say supports their contention that the Cys41-Cys44 equivalent in *M. tuberculosis* & *C. glutamicum* is the redox trigger. Since this study has not been peer-reviewed (or indeed submitted) it is impossible for us to comment on the veracity of these data. We believe we have now addressed all the comments of the referees.